# TruthRL: Incentivizing Truthful LLMs via Reinforcement Learning

Zhepei Wei[1]  Xiao Yang[2]  Kai Sun[2]  Jiaqi Wang[2]  Rulin Shao[3]  Jingxiang Chen[2]
Mohammad Kachuee[2]  Teja Gollapudi[2]  Yiwei Liao[2]  Nicolas Scheffer[2]  Rakesh Wanga[2]
Anuj Kumar[2]  Yu Meng[1]  Wen-tau Yih[4]  Xin Luna Dong[2]

## Abstract

While large language models (LLMs) have demonstrated strong performance on factoid question answering, they are still prone to hallucination and untruthful responses, particularly when tasks demand information outside their parametric knowledge. Indeed, truthfulness requires more than accuracy—models must also recognize uncertainty and abstain when unsure to avoid hallucinations. This presents a fundamental challenge for existing methods: approaches that optimize for accuracy often amplify hallucinations, while those that encourage abstention can become overly conservative, sacrificing correct answers. Both extremes ultimately compromise truthfulness. In this work, we present TruthRL, a general reinforcement learning (RL) framework that directly optimizes the truthfulness of LLMs. Specifically, we implement TruthRL using GRPO with a simple yet effective ternary reward that distinguishes correct answers, hallucinations, and abstentions. It incentivizes models to reduce hallucinations not only by providing correct responses, but also by enabling abstention when uncertain, thereby improving truthfulness. Extensive experiments across four knowledge-intensive benchmarks show that TruthRL significantly reduces hallucinations (*e.g.*, 43.5% → 19.4%) and improves truthfulness (*e.g.*, 5.3% → 37.2%), with consistent gains across various backbone models. Analysis shows that the improvement of TruthRL arises from enhanced capability of LLMs to recognize their knowledge boundary, hence avoiding being overly conservative as the baselines are.[1]

[1]University of Virginia [2]Meta Reality Labs [3]University of Washington [4]FAIR at Meta. Correspondence to: Zhepei Wei <zhepei.wei@virginia.edu>.

*Proceedings of the 43rd International Conference on Machine Learning*, Seoul, South Korea. PMLR 306, 2026. Copyright 2026 by the author(s).

[1]Code: https://github.com/facebookresearch/TruthRL.

## 1. Introduction

While large language models (LLMs) have demonstrated remarkable abilities in generating factual responses (Brown et al., 2020; Touvron et al., 2023; Team et al., 2023), they tend to produce plausible but factually incorrect statements rather than acknowledge uncertainty when encountering questions beyond their knowledge (Xu et al., 2024c; Zhang et al., 2023). This hallucination behavior is especially concerning in high-stakes domains (*e.g.*, law, medicine) where inaccurate outputs can cause severe consequences (Singhal et al., 2023; Xiao et al., 2021; Xiong et al., 2024)—In such scenarios, the model's capability to admit "I don't know" can be just as critical as providing correct information, and a truthful LLM should avoid hallucinations as much as possible. From this perspective, a model that answers fewer questions correctly while reliably abstaining when uncertain is far more *trustworthy* than a higher-accuracy model that frequently fabricates plausible but incorrect answers. In high-stakes domains, such misleading answers risk doing far more harm than abstention. This underscores that *factual accuracy alone does not necessarily guarantee truthfulness*.

There has been a line of research aiming to teach LLMs to admit uncertainty (Cheng et al., 2024; Yang et al., 2024b; Huang et al., 2025). Recent works such as R-Tuning (Zhang et al., 2024a) train the model on unanswerable questions with "I don't know" as the ground-truth label (Song et al., 2025). However, such methods require non-trivial annotation on model-specific datasets, leading to limited generalization or overly conservative behavior (*e.g.*, abstaining even when the model has sufficient knowledge). On the other hand, lots of research efforts have sought to mitigate hallucinations by expanding the model's knowledge scope, either by updating its parametric knowledge through fine-tuning or by incorporating external information via retrieval-augmented generation (RAG) (Kasai et al., 2024; Yang et al., 2024a). However, the retrieved documents in RAG can be noisy or even contain factually incorrect content, potentially misleading the model and posing additional challenges (Wei et al., 2025). Meanwhile, fine-tuning methods typically improve accuracy but can also reinforce hallucinations, particularly when the model is uncertain (Kang et al., 2025). In fact, such accuracy-driven methods inherently motivate

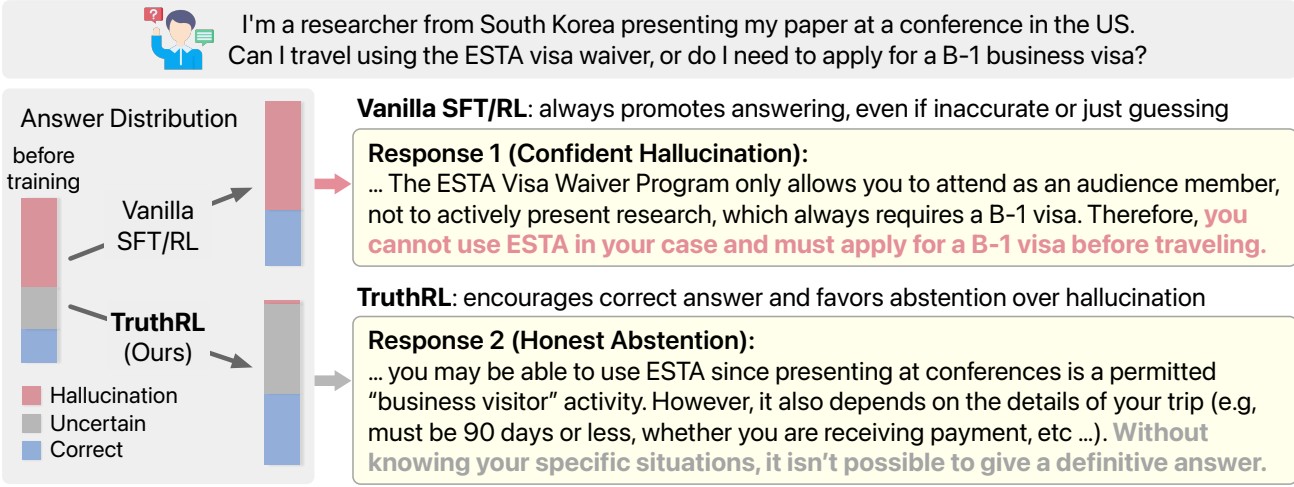

*Figure 1.* Comparison between vanilla supervised fine-tuning (SFT), reinforcement learning (RL), and TruthRL. In vanilla SFT/RL, the model is optimized solely for accuracy, implicitly rewarding hallucinations over abstentions and thus always attempting to answer or guess, which ultimately compromises truthfulness. In contrast, TruthRL not only rewards correct answers, but explicitly penalizes hallucinations, and treats abstentions neutrally, thereby leading to greater truthfulness.

LLMs to guess rather than abstain from answering when unsure, since the expected incentive for guessing an answer is always higher than that from abstention by design (Kalai et al., 2025). As a result, existing approaches remain deficient in training truthful LLMs that can both provide accurate answers and acknowledge uncertainty.

In light of this, we argue that a more aligned learning objective is needed for developing truthful LLMs—one that explicitly incentivizes models not only to maximize correct responses, but also to appropriately abstain from answering when being uncertain. In this work, we introduce TruthRL, a general reinforcement learning (RL) framework designed to directly optimize truthfulness rather than accuracy alone. As illustrated in Figure 1, unlike accuracy-driven methods such as vanilla SFT or RL, which implicitly favor hallucinations over abstentions by encouraging the model to always provide an answer to maximize accuracy, our method introduces a truthfulness-driven ternary reward design that explicitly rewards correct answers, penalizes hallucinations, and treats abstentions as neutral. This design encourages the model to generate correct responses when possible, but more importantly, to properly abstain rather than wildly guessing. Specifically, we implement TruthRL with GRPO (Shao et al., 2024), and our findings show that this simple yet principled reward formulation yields substantial gains in truthfulness. Experiments demonstrate that our method improves the truthfulness of LLMs not only by converting hallucinations into abstentions, but also by promoting more accurate responses, particularly in retrieval-augmented settings where the model has access to additional information. Notably, the increase in abstentions arises not from over-conservatism but from a genuine recognition of the knowledge boundary: TruthRL abstains most often, whereas the baseline tends to hallucinate when knowledge is insufficient.

In summary, our findings advocate a shift from accuracy-driven to truthfulness-driven methods for developing LLMs.

The main contributions of this work are as follows: (1) We propose TruthRL, a general RL framework that directly optimizes truthfulness through a simple yet principled reward design. (2) We demonstrate the effectiveness of TruthRL across multiple knowledge-intensive benchmarks in both retrieval and non-retrieval settings, significantly reducing hallucinations (*e.g.*, 43.5% → 19.4%) and improving truthfulness (*e.g.*, 5.3% → 37.2%). (3) Extensive ablation studies and analyses confirm that LLMs trained with TruthRL are effective at recognizing their knowledge boundaries, robust to hallucination-inducing questions, and more confident in providing correct answers, while maintaining a significantly lower hallucination rate.

## 2. Preliminaries

### 2.1. Problem Formulation

In contrast to the traditional method that optimizes for accuracy only (Kalai et al., 2025), we choose to optimize for *truthfulness* and designed a multi-dimensional objective. Let $\mathcal{D} = \{(x_i, y_i)\}_{i=1}^N$ denote the problem set. For a model $f_\theta$, we evaluate its predictions $\hat{y}_i = f_\theta(x_i)$ and compute (i) accuracy ($\mathrm{Acc}$), the fraction of questions answered correctly; (ii) uncertainty rate ($\mathrm{Unc}$), the fraction of questions where the model abstains (e.g., answers "I don't know"); and (iii) hallucination rate ($\mathrm{Hall}$), the fraction of responses that are factually incorrect. Following standard practices (Yang et al., 2024a; Kachuee et al., 2025; Huang et al., 2025), we define the truthfulness score as a weighted combination: $\mathrm{Truthfulness} = w_1 \cdot \mathrm{Acc} + w_2 \cdot \mathrm{Unc} - w_3 \cdot \mathrm{Hall}$, where $w_1, w_2, w_3$ control the desired behavior among the

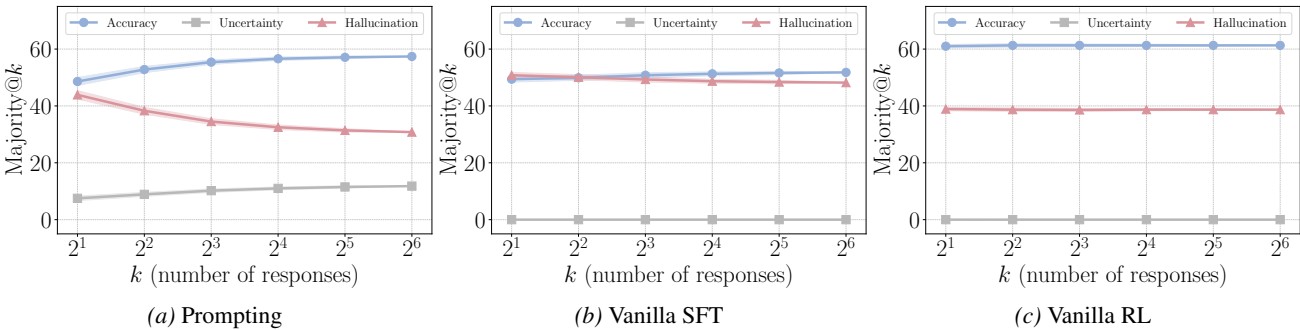

*Figure 2.* Scaling curve of prompting and vanilla SFT/RL methods on the CRAG benchmark, using Llama3.1-8B-Instruct as the backbone. Before training, the model shows strong potential in majority@k scaling, with reduced hallucination and improved accuracy and abstentions as the number of responses increases. However, despite their slightly improved accuracy, vanilla SFT and RL diminish this potential and lead to much higher hallucinations, underscoring their limitations and the need for a more truthful training paradigm.

three dimensions — $w_1, w_3 \geq 0$ meaning that accuracy is always rewarded and hallucination is always penalized, while $w_2$ is determined by deployment needs. By default, we set $w_1 = 1, w_2 = 0, w_3 = 1$. Our objective is to design training methods that maximize the expected truthfulness score, i.e., $\max_\theta \mathbb{E}_\mathcal{D}[\text{Truthfulness}(f_\theta)]$. This formulation captures the core idea of truthfulness: unlike an accuracy-focused setup that only cares about correctness, our problem formulation favors models that maximize correct answers, appropriately abstain when uncertain, and minimize hallucinations.

### 2.2. Vanilla Fine-tuning Methods

**Supervised fine-tuning (SFT).** We train the LLM using the standard SFT objective, which aims to maximize the likelihood of producing the ground-truth response given an input: $\mathcal{L}_{\text{SFT}}(\theta) = -\mathbb{E}_{(x,y)\sim\mathcal{D}} \sum \log p(y|x; \theta)$, where $(x, y)$ is the input-output pair and $\theta$ denotes the parameters. While typically effective for improving accuracy, SFT tends to memorize the training data and has limited generalizability (Chu et al., 2025). Moreover, the model is trained to always provide an answer, even when unsure, which inevitably encourages hallucinations (Kalai et al., 2025).

**Reinforcement learning (RL).** Traditional RL methods optimize the LLM using accuracy-based binary reward signals, provided by a verifier that determines whether a prediction is correct (Guo et al., 2025). Although RL typically achieves better generalization than SFT by eliminating direct supervision with ground-truth answers, vanilla RL is not explicitly designed to recognize uncertainty or abstain when appropriate. As a result, it may substantially increase correctness but still fails to prevent hallucinations (Kang et al., 2025), as also observed in our preliminary findings.

### 2.3. Preliminary Findings

To demonstrate the limitations of vanilla fine-tuning methods (introduced in Section 2.2), we first examine whether

standard accuracy-driven strategies, including prompting, supervised fine-tuning (SFT), and vanilla RL, can reliably balance accuracy and abstention. In Figure 2, we report majority@$k$ results on the CRAG benchmark using Llama3.1-8B-Instruct as the backbone, where majority@$k$ samples $k$ responses and selects the final answer by majority voting.

The prompting baseline demonstrates that increasing the number of sampled responses consistently reduces hallucination with improved accuracy and the abstention rate. This suggests that even without fine-tuning, the base model already has strong potential in achieving higher truthfulness.

In contrast, despite their improvements in accuracy, these methods almost completely suppress abstention behavior (i.e., maintaining a near-zero uncertainty rate) and provide only a limited reduction in hallucinations—or even an increased hallucination rate compared to the baseline when $k$ is large. These results reveal the limitations of vanilla fine-tuning methods that focus solely on accuracy: they not only fail to address the truthfulness problem but also diminish the model's inherent capacity to express uncertainty, underscoring the need for truthful training approaches.

## 3. Methodology

In this section, we first establish strong fine-tuning baselines that can express uncertainty while maintaining accuracy using a knowledge boundary probing mechanism. We then elaborate on the design of TruthRL, and describe how it can operate with or without knowledge-boundary information.

### 3.1. Knowledge Boundary Probing

To enable the model to express uncertainty, we first probe an LLM's knowledge boundaries to identify *out-of-knowledge* (OOK) questions. For each training question, we sample 256 responses with a temperature of 0.6 and top-p of 0.9, and the question is marked as OOK if none of the responses is correct. These questions are then relabeled with "I don't

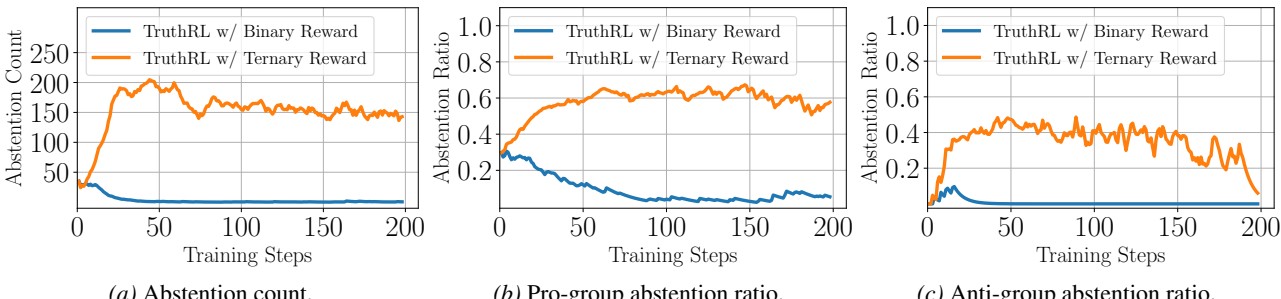

*(a)* Abstention count.        *(b)* Pro-group abstention ratio.        *(c)* Anti-group abstention ratio.

*Figure 3.* Abstention dynamic of TruthRL under different reward designs, with Llama3.1-8B-Instruct as the backbone model.

know" as the ground-truth answer and used to train the model with the standard SFT objective. Similar approaches have been explored in prior works (Zhang et al., 2024a; Yang et al., 2024b; Song et al., 2025), where samples with uncertain labels are incorporated into SFT training using various data construction strategies. We refer to this baseline as R-Tuning (Zhang et al., 2024a).

Additionally, we extend the idea of rejection sampling fine-tuning (RFT) (Yuan et al., 2023) with uncertain responses. Rather than directly learning the ground-truth answers, RFT trains the model on reasoning traces generated by the model itself. In this baseline, we prompt the model to generate multiple reasoning traces for each question and select the trace that concludes with "I don't know" as the label for OOK questions, whereas for non-OOK questions, we select the trace that leads to the correct answer.

### 3.2. TruthRL: Incentivizing Truthfulness via RL

We implement TruthRL using GRPO (Shao et al., 2024), an online RL method that optimizes the following objective:

$$\mathcal{L}_{\text{GRPO}}(\theta) = -\mathbb{E}_{x \sim \mathcal{D}, \{y_i\}_{i=1}^{G} \sim \pi_{\theta_{\text{old}}}(\cdot|x)}$$

$$\left[ \frac{1}{G} \sum_{i=1}^{G} \frac{1}{|y_i|} \sum_{t=1}^{|y_i|} \ell_{i,t}(\theta) - \beta \mathbb{D}_{KL}\left(\pi_\theta || \pi_{\text{ref}}\right) \right],$$

$$\ell_{i,t}(\theta) = \min\left( w_{i,t}(\theta)\hat{A}_i, \text{clip}(w_{i,t}(\theta), 1-\epsilon, 1+\epsilon)\hat{A}_i \right),$$

where $\epsilon$ and $\beta$ are hyper-parameters, $G$ is the group size (*i.e.*, the number of sampled responses from the old policy $\pi_{\theta_{\text{old}}}$ for each question $x$), $\pi_{\text{ref}}$ is the reference policy, $w_{i,t}(\theta)$ is the importance ratio, $\hat{A}_i$ is the estimated advantage for response $y_i$, computed using a group of rewards $\{r_1, \ldots, r_G\}$ corresponding to the outputs within each group:

$$\hat{A}_i = \frac{r(x, y_i) - \text{mean}\left(\{r(x, y_j)\}_{j=1}^{G}\right)}{\text{std}\left(\{r(x, y_j)\}_{j=1}^{G}\right)}.$$

**Reward design.** We consider two general types of reward scheme: (1) Binary reward:

$$r_{\text{binary}}(x, y) = \begin{cases} +1, & \text{if } y \text{ is correct}, \\ -1, & \text{otherwise}. \end{cases}$$

and (2) Ternary reward:

$$r_{\text{ternary}}(x, y) = \begin{cases} +1, & \text{if } y \text{ is correct}, \\ 0, & \text{if } y \text{ is uncertain}, \\ -1, & \text{if } y \text{ is incorrect}. \end{cases}$$

As introduced above, GRPO computes the advantage of a response by comparing its reward against the mean reward within a sampled group. For example, consider a group consisting of two responses $y_1$ and $y_2$ where $y_1$ expresses *abstention* and $y_2$ contains *hallucination*.

- Under the *binary* reward scheme, both responses receive a negative reward (*i.e.*, $r_{\text{binary}}(x, y_1) = r_{\text{binary}}(x, y_2) = -1$), yielding a zero relative advantage (*i.e.*, $\hat{A}_{\text{binary}}(x, y_1) = \hat{A}_{\text{binary}}(x, y_2)$)—thus the policy update conflates hallucination with abstention.

- Under the *ternary* reward scheme, $y_1$ receives a neutral reward while $y_2$ receives a negative reward (*i.e.*, $r_{\text{ternary}}(x, y_1) = 0$, $r_{\text{ternary}}(x, y_2) = -1$), resulting in a larger advantage for abstention than for hallucination (*i.e.*, $\hat{A}_{\text{ternary}}(x, y_1) > \hat{A}_{\text{ternary}}(x, y_2)$). This encourages the models to abstain rather than hallucinate when they lack the knowledge to make accurate predictions.

This exemplifies how the ternary reward inherently better distinguishes abstention from hallucination with the relative advantage estimation of GRPO. Empirically, we also observe clear behavioral divergences between models trained with binary and ternary rewards. As shown in Figure 3a, the binary scheme quickly suppresses abstention behavior to nearly zero, whereas the ternary scheme maintains a meaningful level of abstention. To further understand this effect, we categorize sampling groups into two types: (1) Pro-abstention groups where the model is encouraged to abstain (*i.e.*, groups containing only incorrect and abstention responses). In this group, the abstention ratio consistently increases across training under the ternary reward (Figure 3b). (2) Anti-abstention groups where abstention is penalized (*i.e.*, groups including abstention and at least one correct response). In this group, ternary reward drives the abstention ratio to decrease over training (Figure 3c), as consistent

*Table 1.* Comparison of TruthRL and baselines across four knowledge-intensive benchmarks under with and without retrieval settings. We report the truthfulness score (T), hallucination rate (H), and accuracy (A). The best truthfulness scores are highlighted in **bold**.

| Method | CRAG | | | NQ | | | HotpotQA | | | MuSiQue | | | Average | | |
|---|---|---|---|---|---|---|---|---|---|---|---|---|---|---|---|
| | **T** (↑) | H (↓) | A (↑) | **T** (↑) | H (↓) | A (↑) | **T** (↑) | H (↓) | A (↑) | **T** (↑) | H (↓) | A (↑) | **T** (↑) | H (↓) | A (↑) |
| *Without Retrieval* | | | | | | | | | | | | | | | |
| **Qwen2.5-7B-Inst** | | | | | | | | | | | | | | | |
| Prompting | -17.4 | 48.1 | 30.6 | -32.4 | 60.1 | 27.7 | -36.0 | 60.7 | 24.6 | -68.8 | 76.7 | 7.9 | -38.7 | 61.4 | 22.7 |
| SFT | -51.5 | 75.7 | 24.3 | -49.4 | 74.7 | 25.3 | -46.7 | 73.4 | 26.6 | -81.8 | 90.9 | 9.1 | -57.4 | 78.7 | 21.3 |
| RFT | -16.8 | 46.7 | 29.9 | -20.8 | 49.6 | 28.9 | -19.1 | 46.5 | 27.4 | -41.8 | 50.7 | 8.9 | -24.6 | 48.4 | 23.8 |
| R-Tuning | -7.5 | 21.9 | 14.5 | **-0.9** | 12.6 | 11.7 | 3.3 | 8.4 | 11.7 | **-0.7** | 2.1 | 1.4 | -1.5 | 11.3 | 9.8 |
| RLHF | -17.4 | 45.7 | 28.3 | -31.8 | 60.8 | 29.0 | -39.8 | 63.3 | 23.5 | -73.6 | 80.0 | 6.4 | -40.7 | 62.5 | 21.8 |
| RLKF | -6.1 | 28.1 | 22.1 | -15.7 | 37.9 | 22.2 | -18.1 | 36.2 | 18.1 | -50.6 | 54.0 | 3.4 | -22.6 | 39.1 | 16.5 |
| TruthRL_Binary | -29.2 | 64.5 | 35.3 | -35.9 | 67.8 | 31.9 | -31.2 | 65.3 | 34.1 | -71.7 | 84.8 | 13.2 | -42.0 | 70.6 | 28.6 |
| TruthRL | **16.2** | 8.7 | 24.9 | -1.6 | 25.0 | 23.5 | **9.8** | 12.7 | 22.5 | -1.7 | 5.3 | 3.6 | **5.7** | 12.9 | 18.6 |
| **Llama3.1-8B-Inst** | | | | | | | | | | | | | | | |
| Prompting | -4.4 | 44.5 | 40.1 | -5.2 | 49.2 | 43.9 | -19.9 | 53.9 | 34.0 | -54.2 | 64.7 | 10.5 | -20.9 | 53.1 | 32.1 |
| SFT | -42.1 | 71.1 | 28.9 | -38.4 | 69.2 | 30.8 | -38.9 | 69.5 | 30.5 | -81.9 | 90.9 | 9.1 | -50.3 | 75.2 | 24.8 |
| RFT | -7.6 | 48.1 | 40.4 | -11.4 | 51.8 | 40.4 | -23.2 | 57.9 | 34.7 | -58.0 | 69.2 | 11.2 | -25.1 | 56.8 | 31.7 |
| R-Tuning | -13.7 | 39.5 | 25.8 | -16.6 | 42.5 | 25.9 | -3.5 | 26.7 | 23.2 | -20.7 | 25.2 | 4.5 | -13.6 | 33.5 | 19.9 |
| RLHF | 1.4 | 35.3 | 36.7 | -37.5 | 60.8 | 23.3 | -39.4 | 57.3 | 17.9 | -72.9 | 77.8 | 5.0 | -37.1 | 57.8 | 20.7 |
| RLKF | -0.6 | 32.8 | 32.2 | -4.6 | 37.9 | 33.2 | -4.5 | 31.8 | 27.3 | -29.3 | 36.2 | 6.9 | -9.8 | 34.7 | 24.9 |
| TruthRL_Binary | -14.5 | 57.2 | 42.8 | -5.3 | 52.6 | 47.4 | -19.6 | 59.8 | 40.2 | -67.2 | 83.6 | 16.4 | -26.7 | 63.3 | 36.7 |
| TruthRL | **22.4** | 16.3 | 38.7 | **12.9** | 30.9 | 43.8 | **14.3** | 18.9 | 33.2 | **-7.7** | 16.0 | 8.2 | **10.5** | 20.5 | 31.0 |
| *With Retrieval* | | | | | | | | | | | | | | | |
| **Qwen2.5-7B-Inst** | | | | | | | | | | | | | | | |
| Prompting | 10.6 | 38.4 | 49.0 | 9.0 | 41.1 | 50.1 | 0.2 | 43.6 | 43.8 | -51.3 | 62.8 | 11.5 | -7.9 | 46.5 | 38.6 |
| SFT | -2.3 | 51.2 | 48.8 | 0.3 | 49.9 | 50.1 | -2.4 | 51.2 | 48.8 | -68.2 | 84.1 | 15.9 | -18.2 | 59.1 | 40.9 |
| RFT | 22.6 | 31.4 | 54.0 | 18.4 | 32.1 | 50.5 | 23.4 | 23.3 | 46.6 | -20.6 | 33.8 | 13.2 | 11.0 | 30.2 | 41.1 |
| R-Tuning | 13.4 | 35.0 | 48.4 | 4.3 | 44.5 | 48.8 | 13.8 | 30.3 | 44.1 | -23.0 | 32.5 | 9.4 | 2.1 | 35.6 | 37.7 |
| RLHF | 16.5 | 35.9 | 52.4 | 10.9 | 41.6 | 52.5 | 9.5 | 39.4 | 48.8 | -52.1 | 65.6 | 13.5 | -3.8 | 45.6 | 41.8 |
| RLKF | 25.2 | 21.9 | 47.1 | 18.4 | 21.9 | 40.4 | 20.4 | 11.9 | 32.3 | -13.8 | 19.1 | 5.3 | 12.6 | 18.7 | 31.3 |
| TruthRL_Binary | 8.4 | 45.3 | 53.7 | 11.8 | 43.9 | 55.7 | 20.1 | 39.1 | 59.2 | -49.4 | 72.2 | 22.8 | -2.3 | 50.1 | 47.9 |
| TruthRL | **33.1** | 17.3 | 50.4 | **26.4** | 21.2 | 47.6 | **33.3** | 10.7 | 43.9 | **-0.6** | 9.0 | 8.4 | **23.1** | 14.6 | 37.6 |
| **Llama3.1-8B-Inst** | | | | | | | | | | | | | | | |
| Prompting | 5.3 | 43.5 | 48.8 | -5.8 | 50.7 | 44.9 | -4.4 | 49.0 | 44.6 | -60.5 | 73.0 | 12.5 | -16.4 | 54.1 | 37.7 |
| SFT | 1.4 | 49.3 | 50.7 | 1.6 | 49.2 | 50.8 | -4.3 | 52.1 | 47.9 | -69.8 | 84.9 | 15.1 | -17.8 | 58.9 | 41.1 |
| RFT | -3.7 | 48.8 | 45.1 | -4.7 | 50.4 | 45.7 | 1.1 | 45.8 | 46.9 | -55.7 | 68.8 | 13.1 | -15.8 | 53.5 | 37.7 |
| R-Tuning | 15.2 | 33.1 | 48.4 | 2.1 | 47.5 | 49.6 | 1.7 | 46.2 | 47.9 | -53.9 | 68.3 | 14.4 | -8.7 | 48.8 | 40.1 |
| RLHF | 13.1 | 39.7 | 52.7 | 8.2 | 44.2 | 52.4 | 9.6 | 41.9 | 51.6 | -57.4 | 72.4 | 15.0 | -6.6 | 49.6 | 42.9 |
| RLKF | 18.9 | 30.6 | 49.5 | 10.2 | 34.9 | 45.0 | 10.8 | 32.7 | 43.5 | -33.5 | 43.4 | 9.9 | 1.6 | 35.4 | 37.0 |
| TruthRL_Binary | 20.8 | 39.5 | 60.3 | 19.0 | 40.5 | 59.5 | 25.9 | 37.0 | 62.9 | -47.6 | 73.7 | 26.1 | 4.5 | 47.7 | 52.2 |
| TruthRL | **37.2** | 19.4 | 56.6 | **28.8** | 24.9 | 53.7 | **37.4** | 14.9 | 52.3 | **-0.9** | 15.9 | 15.0 | **25.6** | 18.8 | 44.4 |

with the intended behavior. This shows that ternary rewards under GRPO adaptively generate positive or negative credits for abstention, ultimately resulting in different abstention dynamics that cannot be reproduced by binary rewards.

On top of the above default design of TruthRL, we further consider two exploratory enhancements. First, a *knowledge-enhanced variant* treats abstention as positive when the model is deemed to lack knowledge. For out-of-knowledge (OOK) questions, it assigns $+1$ to uncertain responses and $-1$ to other responses; for non-OOK questions, it assigns

$+1$ to correct answers, $-1$ for incorrect answers, and $0$ to abstentions if applied with the ternary reward. Second, a *reasoning-enhanced variant* builds on the above outcome-based reward by incorporating additional reward signals that evaluate the quality of the model's reasoning process. Interestingly, our analysis provided in Section 4.4 and Appendix A suggests that TruthRL with a simple ternary reward scheme generally works better than the variants with binary scheme or more complicated designs, demonstrating the simplicity and effectiveness of TruthRL.

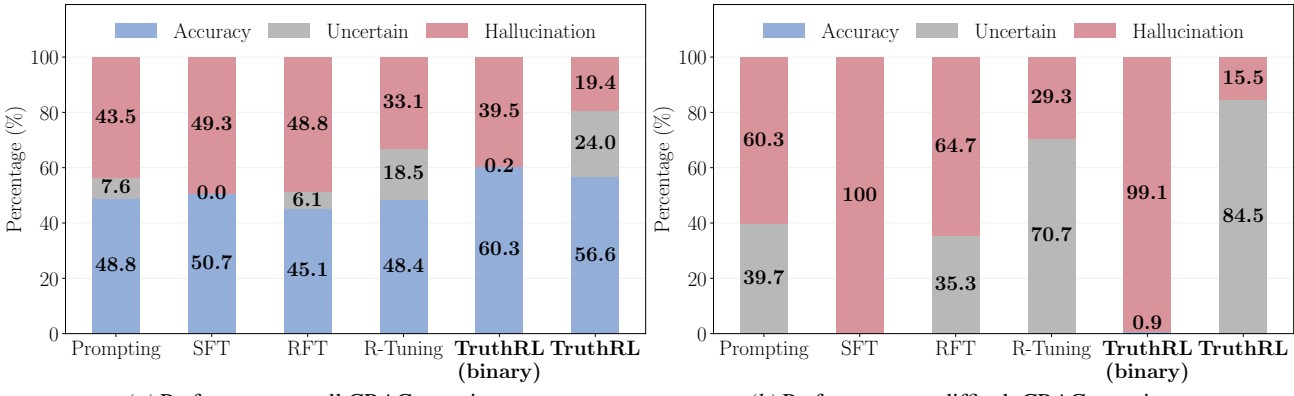

*(a) Performance on all CRAG questions.*      *(b) Performance on difficult CRAG questions.*

*Figure 4.* Performance decomposed to accuracy (blue), hallucination (red), and uncertainty (gray). Compared to baselines, TruthRL achieves the highest overall accuracy and the lowest hallucination. On difficult questions where almost no method can provide correct answers, TruthRL produces minimal hallucinations while other methods hallucinate heavily, demonstrating its improved capability in recognizing knowledge boundaries.

## 4. Experiment

### 4.1. Experimental Setting

**Datasets and Evaluation Metrics.** We conduct experiments on four knowledge-intensive benchmarks, under with and without retrieval setups: CRAG (Yang et al., 2024a), NaturalQuestions (NQ) (Kwiatkowski et al., 2019), HotpotQA (Yang et al., 2018), and MuSiQue (Trivedi et al., 2022). Models are trained on CRAG and evaluated across all four datasets. The primary evaluation metric is *truthfulness* score, with *hallucination* rate and *accuracy* reported as auxiliary metrics.

**Models and Baselines.** We compare TruthRL with a wide range of baselines, including prompting, vanilla SFT, and two knowledge-enhanced SFT, namely RFT and R-Tuning (Section 3). We also include two representative RL-based baselines, RLHF (Ouyang et al., 2022) and RLKF (Xu et al., 2024a). By default, TruthRL is implemented using the ternary reward scheme without any enhancements. We instantiate the above methods using Llama3.1-8B-Instruct (Dubey et al., 2024) and Qwen2.5-7B-Instruct (Qwen et al., 2025) as backbone models. More implementation details are provided in Appendix C and D.

### 4.2. Main Result

**Vanilla SFT increases both accuracy and hallucination, while knowledge-enhanced SFTs effectively reduce hallucination with little loss or even a meaningful gain in accuracy.** As shown in Table 1, compared to the prompting baseline, vanilla SFT substantially increases hallucination rates, showing that simply optimizing for accuracy can inadvertently encourage incorrect answers. This effect is particularly pronounced in the no-retrieval setting, which is likely because using SFT for ground-truth answers that the model does not know encourages the model to generate content beyond its knowledge, thereby promoting hallucinations.

In contrast, knowledge-enhanced SFT methods (*i.e.*, RFT, R-Tuning) achieve much lower hallucination with little to no compromise in accuracy, and can even improve accuracy when sufficient information is provided through retrieval, demonstrating the benefit of explicitly modeling uncertainty.

**TruthRL consistently outperforms baselines in terms of truthfulness, with significantly reduced hallucination and increased accuracy, particularly in the retrieval setup.** Access to external information consistently improves performance for all methods, highlighting the effectiveness of RAG in mitigating hallucinations. On CRAG, TruthRL reduces hallucination for Llama3.1-8B-Instruct by 24.1% and achieves an absolute gain of 31.9% in truthfulness compared to the prompting baseline under the retrieval setup. While knowledge-enhanced SFT (*i.e.*, RFT, R-Tuning) and RL-based methods (*i.e.*, RLHF, RLKF) substantially improve truthfulness over vanilla SFT and prompting baselines, they still struggle to balance hallucination and accuracy. TruthRL_Binary is a variant of our method that uses a binary reward, achieving the highest accuracy but also exhibiting a high hallucination rate and losing the ability of abstention. In contrast, TruthRL reduces hallucinations and also improves accuracy and appropriate abstentions, yielding the highest overall truthfulness.

### 4.3. TruthRL improves LLMs in recognizing their knowledge boundaries

**TruthRL enables LLMs to abstain from answering only when they genuinely lack knowledge.** Figure 4 breaks down performance on the CRAG benchmark under the retrieval setup, evaluating both the full test set and a challenging subset, using Llama3.1-8B-Instruct as the backbone model. On the full set (Figure 4a), compared to the prompting method, fine-tuning baselines either achieve improved accuracy with almost zero uncertainty rate (*e.g.*, SFT, TruthRL_Binary) or sacrifice accuracy to allow abstention (*e.g.*,

*Table 2.* Ablation study on reward design of TruthRL.

| TruthRL | CRAG T (↑) | CRAG H (↓) | NQ T (↑) | NQ H (↓) | HotpotQA T (↑) | HotpotQA H (↓) | MuSiQue T (↑) | MuSiQue H (↓) | Average T (↑) | Average H (↓) |
|---|---|---|---|---|---|---|---|---|---|---|
| with binary reward | 20.8 | 39.5 | 19.0 | 40.5 | 25.9 | 37.0 | -47.6 | 73.7 | 4.5 | 47.7 |
| + knowledge-enhanced | 27.4 | 30.6 | 19.2 | 38.2 | 28.9 | 32.0 | -28.9 | 52.3 | 11.7 | 38.3 |
| with ternary reward | **37.2** | **19.4** | **28.8** | **24.9** | **37.4** | 14.9 | **-0.9** | 15.9 | **25.6** | **18.8** |
| + knowledge-enhanced | 32.7 | 21.9 | 27.2 | 25.3 | 35.1 | **13.4** | -2.3 | **15.0** | 23.2 | 18.9 |

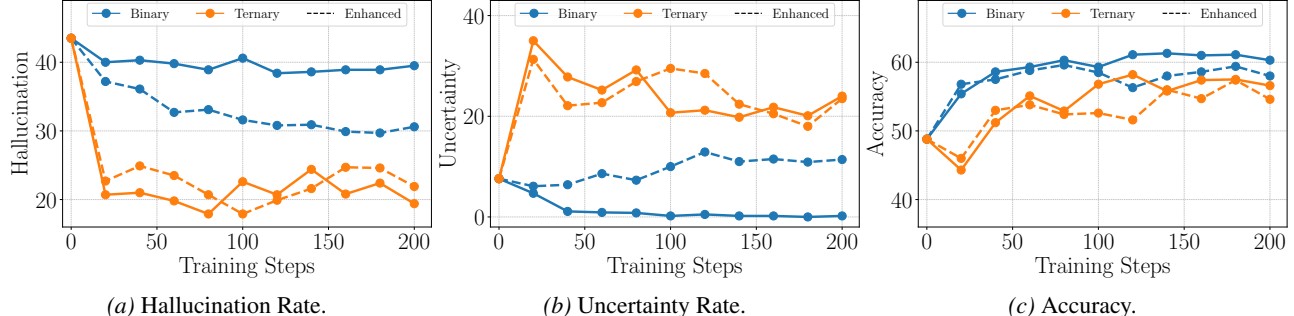

*(a) Hallucination Rate.*      *(b) Uncertainty Rate.*      *(c) Accuracy.*

*Figure 5.* Learning dynamics of TruthRL under different reward designs.

*Table 3.* Performance on hallucination-baiting questions.

| Method | T (↑) | H (↓) | U (−) |
|---|---|---|---|
| Prompting | 9.7 | 39.8 | 10.7 |
| SFT | 3 | 48.5 | 0 |
| RFT | 12.7 | 38.8 | 9.7 |
| R-Tuning | 6.8 | 43.7 | 5.8 |
| TruthRL | **52.4** | **16.5** | 14.6 |

RFT, R-Tuning). In contrast, TruthRL achieves the lowest hallucination rate while maintaining competitive accuracy and the highest uncertainty rate among all baselines.When evaluating on the difficult questions (Figure 4b), where almost no method provides correct answers, all baselines hallucinate heavily—models that achieve high overall accuracy can even hallucinate nearly 100% on these challenging questions (*e.g.*, SFT, TruthRL$_{Binary}$). In contrast, TruthRL produces minimal hallucinations (15.5%) while generating uncertain responses for most cases (84.5%), demonstrating an improved ability to recognize its knowledge boundaries.

**TruthRL is robust to halluciantion-baiting questions.** We evaluate different methods on the comparison-type questions from CRAG, where candidate answers are explicitly provided in the input (*e.g.*, "Which is larger, A or B?"). Such multichoice–like questions are known to be prone to inducing hallucinations (Kang et al., 2025). As shown in Table 3, baselines exhibit high hallucination rates and limited truthfulness scores. Albeit the promising results of knowledge-enhanced baselines in Table 1, they still suffer substantial hallucinations on these hallucination-baiting questions, highlighting their vulnerability. In contrast, TruthRL achieves the highest truthfulness score while maintaining the lowest

hallucination rate, further confirming its effectiveness.

### 4.4. Ablation Study

**Binary reward design excels in accuracy but is limited in truthfulness, while ternary reward achieves the best truthfulness score with competitive accuracy.** As shown in Table 2, the binary reward that only distinguishes correct vs. incorrect answers strongly increases accuracy, but drives the model towards the elimination of abstentions. Augmenting binary reward with knowledge-enhanced signals partially alleviates this issue, improving abstention rates while at the cost of compromised accuracy, as also evidenced in the learning dynamics (Figure 5). Interestingly, knowledge enhancement on top of ternary reward scheme does not lead to improvement, which we attribute to the static nature of knowledge probing (Section 3), as it fails to capture the model's evolving knowledge boundary during training.

**Justification of neutral abstention reward in TruthRL.** As shown in Table 5, we vary the abstention reward during training by setting it to negative (−0.5), neutral (0), and positive (0.5). Assigning either a negative or positive reward leads to suboptimal behavior, as penalizing abstention increases hallucination while rewarding it encourages excessive abstention. In contrast, the neutral reward achieves the best balance, substantially improving truthfulness.

### 4.5. Analysis

Due to space limit, we present additional exploratory and scaling analyses in Appendix A, including the impact of different RL paradigms, verifier selection, model scales, and auxiliary reward designs.

*Table 4.* Evaluation on CRAG under different judges.

| Method | Llama3.3-70B-Inst | | Qwen2.5-72B-Inst | | Gemma3-27B-Inst | | Average | |
|---|---|---|---|---|---|---|---|---|
| | **T** ($\uparrow$) | **H** ($\downarrow$) | **T** ($\uparrow$) | **H** ($\downarrow$) | **T** ($\uparrow$) | **H** ($\downarrow$) | **T** ($\uparrow$) | **H** ($\downarrow$) |
| Prompting | 5.3 | 43.5 | 1.9 | 45.3 | 6.5 | 42.9 | 4.6 | 43.9 |
| SFT | 1.4 | 49.3 | 1.7 | 49.1 | 6.7 | 46.7 | 3.3 | 48.4 |
| RFT | -3.7 | 48.8 | -5.0 | 49.5 | -3.1 | 48.5 | -3.9 | 48.9 |
| R-Tuning | 15.2 | 33.1 | 14.9 | 33.3 | 18.0 | 31.7 | 16.0 | 32.7 |
| TruthRL | **37.2** | **19.4** | **35.6** | **20.2** | **39.7** | **18.2** | **37.5** | **19.3** |

*Table 5.* Training TruthRL with different abstention rewards.

| TruthRL | **T** ($\uparrow$) | **H** ($\downarrow$) | **U** ($-$) |
|---|---|---|---|
| Negative ($-0.5$) | 30.8 | 26.9 | 15.4 |
| Positive ($0.5$) | 34.7 | **15.8** | 33.7 |
| Neutral ($0$) | **37.2** | 19.4 | 24.0 |

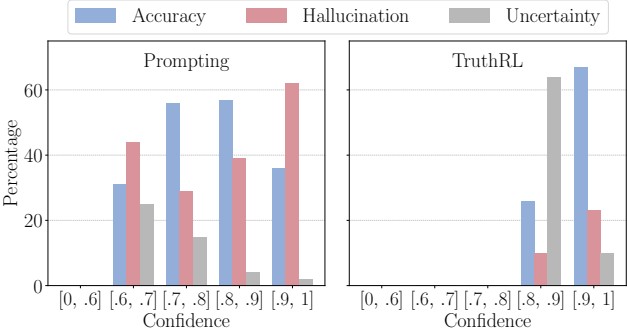

*Figure 6.* Model behaviors under different output confidence.

*Table 6.* Evaluating truthfulness with different abstention weights.

| Method | **T** ($w_2 = -0.5$) | **T** ($w_2 = 0$) | **T** ($w_2 = 0.5$) |
|---|---|---|---|
| Prompting | 1.5 | 5.3 | 9.2 |
| SFT | 1.4 | 1.4 | 1.4 |
| RFT | -6.8 | -3.7 | -0.7 |
| R-Tuning | 5.9 | 15.2 | 24.5 |
| **TruthRL** | **25.2** | **37.2** | **49.2** |

*Table 7.* Evaluation on the GSM8K benchmark, a mathematical reasoning benchmark outside the knowledge-intensive QA training domain. Compared to baseline methods, TruthRL achieves the highest truthfulness and accuracy with a low hallucination rate, demonstrating strong generalization beyond the training domain.

| Method | **T** ($\uparrow$) | **H** ($\downarrow$) | **U** | **A** ($\uparrow$) |
|---|---|---|---|---|
| Prompting | 66.7 | 16.3 | 0.7 | 83.0 |
| SFT | -67.4 | 83.7 | 0.0 | 16.3 |
| RFT | 59.8 | 19.6 | 0.9 | 79.5 |
| R-Tuning | -4.0 | **6.4** | 91.3 | 2.4 |
| **TruthRL** | **71.9** | 12.7 | 2.7 | **84.6** |

**TruthRL is more confident in giving correct answers and abstaining, while the hallucination rate is significantly lower.** As shown in Figure 6, we group the model outputs based on their confidence intervals. Even before fine-tuning, Llama3.1-8B-Instruct already exhibits high confidence in its predictions. However, a large portion of outputs in each confidence interval are hallucinations. Moreover, the uncertainty rate decreases as confidence increases, indicating that the model tends to provide an answer rather than abstain when its confidence is high. In contrast, TruthRL further increases the confidence of model outputs, it not only improves accuracy but also significantly reduces overconfident hallucinations. This indicates that our method produces responses that are not only more accurate but also better aligned with their confidence.

**TruthRL is robust across different LLM judges.** Table 4 reports results on CRAG under three distinct high-capacity evaluators: Llama3.3-70B-Instruct, Qwen2.5-72B-Instruct, and Gemma3-27B-Instruct. While absolute scores vary slightly across judges, the relative improvements of TruthRL are consistent: it achieves the lowest hallucination and the highest truthfulness under all evaluators. This robustness suggests that TruthRL learns generalizable behaviors rather

than overfitting to the idiosyncrasies of a single judge.

**TruthRL consistently outperforms baselines under various truthfulness evaluation setups.** By default, we set $w_1 = 1, w_2 = 0, w_3 = 1$ for truthfulness score calculation. That said, the abstention weight in the truthfulness metric may vary depending on application needs. As shown in Table 6, we calculate truthfulness with $w_2 \in \{-0.5, 0, 0.5\}$. TruthRL consistently achieves the highest scores, demonstrating superiority under varying evaluation scenarios.

**TruthRL generalizes well beyond the training domain.** TruthRL generalizes effectively from CRAG training to diverse unseen QA test scenarios, including NQ (single-hop factoid), HotpotQA (multi-hop reasoning), and MuSiQue (multi-step compositional), suggesting that it learns knowledge-boundary recognition rather than dataset-specific patterns. In Table 7, we further evaluate on GSM8K, a mathematical reasoning benchmark outside knowledge-intensive QA, and find that TruthRL preserves reasoning accuracy while reducing hallucinated answers, providing additional evidence of generalization beyond the original QA training domain.

# 5. Related Work

LLMs often generate factually incorrect statements, or hallucinations (Zhang et al., 2023), arising from limited grounding (Shuster et al., 2021) and over-reliance on parametric memory (Petroni et al., 2019). Mitigation strategies include retrieval-augmented generation (Lewis et al., 2020), decoding-based self-correction (Wang et al., 2023), and fine-tuning methods such as SFT (Zhou et al., 2023) and RLHF (Ouyang et al., 2022), though these approaches often fail out-of-distribution (Kirk et al., 2024) and rarely model abstention. Reinforcement learning (Zhu et al., 2025) elicits reasoning for improved accuracy with binary correctness rewards but penalizes abstention (Song et al., 2025). Extensions such as R-Tuning (Zhang et al., 2024a) reduces hallucinations but sacrifices coverage, highlighting the need for reward formulations that balance factual accuracy with calibrated abstention. Due to space limit, we provide an extended discussion in Appendix B.

# 6. Conclusion

We presented TruthRL, a general reinforcement learning framework that directly optimizes LLM truthfulness. Using a simple yet effective ternary reward, TruthRL encourages accurate answers, abstention when uncertain, and reduce hallucinations. Compared to baselines, it achieves up to 43.4% improvement in truthfulness and 40.1% reduction in hallucinations on average across four knowledge-intensive benchmarks, demonstrating the effectiveness of TruthRL.

# Impact Statement

This work focuses on improving the truthfulness of large language models (LLMs), addressing the problem of hallucinations and untruthful responses in knowledge-intensive tasks. TruthRL incentivizes models not only to provide more accurate answers but also to abstain when uncertain, improving reliability and trustworthiness of LLMs. While the general ethical concerns associated with LLMs (*e.g.*, fairness, bias, and potential misuse) still exist, our method contributes positively by promoting more truthful and responsible outputs. We believe this can benefit applications in high-stakes or knowledge-critical domains, where misleading or fabricated information could have serious consequences.

# Acknowledgements

We thank anonymous reviewers for their constructive and insightful comments.

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

# A. Additional Analysis

**LLM-based verifier provides more reliable training signals than rule-based verifier.** When replacing the LLM-based verifier with a simple rule-based verifier (Table 8), the model collapses into overly conservative behavior, abstaining on the vast majority of queries. Although this results in an extremely low hallucination rate, the truthfulness score becomes negative, reflecting the lack of meaningful answers. By contrast, the LLM-based verifier provides fine-grained, context-sensitive signals that better capture partial correctness and nuanced errors, which stabilizes RL training and leads to higher overall performance. This demonstrates that a high-quality verifier is as important as the reward design itself in reinforcement learning for truthfulness.

*Table 8.* Training with rule-based verifier vs. training with model-based verifier.

| TruthRL | T ($\uparrow$) | H ($\downarrow$) |
|---|---|---|
| with rule-based verifier | -3.6 | 3.6 |
| with model-based verifier | 37.2 | 19.4 |

*Table 9.* Performance of TruthRL with different backbones on the CRAG benchmark.

| | Llama3.2-3B-Inst | | Qwen2.5-3B-Inst | | Qwen2.5-7B-Inst | | Llama3.1-8B-Inst | | Qwen2.5-32B-Inst | |
|---|---|---|---|---|---|---|---|---|---|---|
| **Method** | **T** ($\uparrow$) | **H** ($\downarrow$) | **T** ($\uparrow$) | **H** ($\downarrow$) | **T** ($\uparrow$) | **H** ($\downarrow$) | **T** ($\uparrow$) | **H** ($\downarrow$) | **T** ($\uparrow$) | **H** ($\downarrow$) |
| Prompting | 1.9 | 45.1 | -0.3 | 45.4 | 10.6 | 38.4 | 5.3 | 43.5 | 29.1 | 27.1 |
| TruthRL | 27.4 | 21.5 | 21.9 | 16.2 | 33.1 | 17.3 | 37.2 | 19.4 | 40.0 | 18.2 |

**TruthRL consistently improves across model scales.** We further examine the scalability of our method across a spectrum of model sizes, ranging from compact backbones (*e.g.*, Llama3.2-3B, Qwen2.5-3B) to mid/large-scale models (*e.g.*, Qwen2.5-7B, Llama3.1-8B, Qwen2.5-32B). As summarized in Table 9, TruthRL consistently reduces hallucination and boosts truthfulness regardless of the base model size. Interestingly, the relative gain is more pronounced for smaller models, which suffer from higher hallucination rates under prompting. This suggests that our approach not only strengthens already strong models but also helps weaker models develop more reliable uncertainty-awareness. At the large-model end, the improvements on 32B backbones highlight that even highly capable LLMs benefit from reinforcement learning with uncertainty-aware rewards, underscoring the scalability of our method to state-of-the-art systems.

**Beyond outcome reward.** We conduct a reasoning-quality analysis on CRAG using the prompting method with Llama3.1-8B-Instruct, evaluating model responses for both outcome and reasoning quality, resulting in an overall truthfulness score of 5.3% and a reasoning score of 50.2%. Specifically, results show that accurate responses are typically associated with a high reasoning quality score of 92% and uncertain responses exhibit a reasoning score of 0%, while hallucinated responses have a reasoning sore of 12.1%. The findings suggest a strong correlation between response accuracy and reasoning quality. The high reasoning score of accurate responses indicates that the model excels in generating accurate outcomes with promising reasoning. However, the low reasoning scores for uncertain and hallucinated responses highlight the need for quality reasoning. Introducing reasoning rewards could potentially help mitigate these issues, enabling more accurate outcomes with better reasoning.

We explore three heuristic strategies for incorporating the reasoning reward $r_{\text{reason}}$ on top of the outcome reward $r_{\text{outcome}}$: (1) A multiplicative strategy scales the outcome reward by reasoning quality, i.e., $r_{\text{final}} = r_{\text{outcome}} \cdot (1 + r_{\text{reason}})$, which particularly encourages better reasoning when the outcome is correct. (2) An additive strategy treats reasoning as a complementary signal with scaling factor $\lambda$, giving $r_{\text{final}} = r_{\text{outcome}} + \lambda \cdot r_{\text{reason}}$, so that good reasoning can get rewarded even when the outcome reward is moderate. (3) A conditional strategy applies reasoning rewards only if the outcome is correct: $r_{\text{final}} = r_{\text{outcome}} \cdot r_{\text{reason}}$ when $r_{\text{outcome}} = 1$, and $r_{\text{final}} = r_{\text{outcome}}$

*Table 10.* Impact of incorporating reasoning reward into TruthRL on CRAG.

| | Outcome | | Reasoning |
|---|---|---|---|
| **Method** | **T** ($\uparrow$) | **H** ($\downarrow$) | **Score** ($\uparrow$) |
| Prompting | 5.3 | 43.5 | 50.2 |
| TruthRL ($r_{\text{outcome}}$ only) | **37.2** | 19.4 | 56.6 |
| + multiplicative $r_{\text{reason}}$ | 37.0 | 19.4 | 54.7 |
| + additive $r_{\text{reason}}$ | 36.1 | **19.1** | **59.1** |
| + conditional $r_{\text{reason}}$ | 35.6 | 19.3 | 55.1 |

otherwise, enforcing stricter alignment where reasoning quality matters primarily in successful completions. The results in Table 10 indicate that outcome-only rewards implicitly improve reasoning ability, while explicitly optimizing reasoning quality requires non-trivial design to balance multiple objectives. For instance, heuristic designs like additive reasoning rewards can boost reasoning scores but may compromise the outcome, underscoring the need for thoughtful design. We leave this exploration for future work.

*Table 11.* Comparison between Offline RL (DPO), Semi-Online RL (Iterative DPO), and Online RL (TruthRL) across four knowledge-intensive benchmarks. The best and second-best scores are highlighted using **bold** and underline, respectively.

| | CRAG | | NQ | | HotpotQA | | MuSiQue | | Average | |
|---|---|---|---|---|---|---|---|---|---|---|
| **Method** | **T (↑)** | **H (↓)** | **T (↑)** | **H (↓)** | **T (↑)** | **H (↓)** | **T (↑)** | **H (↓)** | **T (↑)** | **H (↓)** |
| DPO | 6.8 | 43.7 | -2.8 | 49.8 | 6.3 | 43.8 | -50.5 | 67.0 | -10.1 | 51.1 |
| Iterative DPO | | | | | | | | | | |
| Iter 1 | 12.9 | 40.9 | 6.7 | 45.5 | 12.7 | 41.0 | -49.7 | 67.4 | -4.4 | 48.7 |
| Iter 2 | 16.3 | 39.8 | 9.3 | 43.4 | 18.5 | 37.5 | -44.4 | 61.9 | -0.1 | 45.7 |
| Iter 3 | 28.0 | 29.5 | 15.0 | 37.1 | 26.5 | 26.5 | -19.0 | 33.7 | 12.6 | 31.7 |
| Iter 4 | 19.0 | 33.9 | 4.3 | 44.5 | 8.1 | 40.1 | -39.5 | 52.7 | -2.0 | 42.8 |
| TruthRL | **37.2** | **19.4** | **28.8** | **24.9** | **37.4** | **14.9** | **-0.9** | **15.9** | **25.6** | **18.8** |

**Online RL outperforms offline and semi-online counterparts.** Table 11 compares different reinforcement learning paradigms using the same backbone model Llama3.1-8B-Instruct on the CRAG benchmark under retrieval setup. We observe that purely offline RL via DPO leads to limited gains: although slightly better than promoting baseline, the truthfulness score remains low, as the fixed dataset constrains the model's ability to adaptively refine its behavior. Semi-online training through iterative DPO provides some remedy by refreshing preference data after each iteration, but the performance is inconsistent: early iterations bring steady improvements, yet excessive iterations (e.g., Iter 4) show regressions, suggesting that repeated offline fine-tuning cannot effectively balance exploration and exploitation. In contrast, our TruthRL with online GRPO achieves the best results across all benchmarks, consistently lowering hallucination while improving truthfulness. This highlights the advantage of learning from online interactions, which enables continuous updates and policy refinement without drifting toward overfitting or degeneration. More implementation details on DPO are provided in Appendix C.

*Table 12.* Comparison with GPT-5 and o3 across four knowledge-intensive benchmarks.

| | CRAG | | NQ | | HotpotQA | | MuSiQue | | Average | |
|---|---|---|---|---|---|---|---|---|---|---|
| **Method** | **T (↑)** | **H (↓)** | **T (↑)** | **H (↓)** | **T (↑)** | **H (↓)** | **T (↑)** | **H (↓)** | **T (↑)** | **H (↓)** |
| OpenAI o3 | 33.2 | 33.1 | 38.1 | 30.8 | **63.6** | 18.0 | 1.0 | 49.0 | 34.0 | 32.7 |
| GPT-5 | **42.3** | 24.7 | **39.8** | 28.0 | 62.3 | **17.3** | **2.9** | 43.3 | **36.8** | 28.3 |
| Llama3.3-70B-Inst | | | | | | | | | | |
| Prompting | 32.1 | 26.4 | 28.1 | 30.8 | 42.5 | 19.6 | -14.2 | 37.7 | 22.1 | 28.6 |
| TruthRL | 40.6 | **17.3** | 30.7 | **26.0** | 48.5 | 17.4 | -0.1 | **23.3** | 29.9 | **21.0** |

**TruthRL achieves competitive performance compared to production-level models like GPT-5.** As shown in Table 12, TruthRL consistently delivers strong truthfulness scores and low hallucination rates across four knowledge-intensive benchmarks. While GPT-5 and OpenAI o3 remain highly capable, TruthRL matches or surpasses them on several datasets. For example, it achieves the lowest hallucination rate and the second-highest truthfulness score on CRAG. Notably, TruthRL substantially improves the backbone model Llama3.3-70B-Instruct to a level comparable with GPT-5 and OpenAI o3, demonstrating its potential to close the gap between open-source LLMs and production-grade models. These results underscore the effectiveness of TruthRL in enhancing truthfulness in knowledge-intensive applications.

## B. Extended Related Work

**LLM Hallucination and Mitigation** A recurring challenge in LLMs is their tendency to generate fluent but factually incorrect statements, commonly termed hallucinations (Ji et al., 2023; Zhang et al., 2023; Xu et al., 2024c; Feng et al., 2024; Zhang et al., 2024c; Bang et al., 2025; Orgad et al., 2025; Luo et al., 2026). Hallucinations arise from several factors, including limited grounding in external knowledge sources (Shuster et al., 2021) and over-reliance on parametric recall (Petroni et al., 2019). This gap between surface-level accuracy and deeper factual correctness becomes especially problematic in high-stakes domains such as law (Xiao et al., 2021) and medicine (Singhal et al., 2023; Xiong et al., 2024), where confidently wrong outputs can mislead users. Several lines of mitigation strategies have been explored (Wen et al., 2025). *Retrieval-augmented* methods ground LLMs in external knowledge bases or search engines to reduce reliance on memory alone (Borgeaud et al., 2022; Izacard et al., 2023; Chen et al., 2025a; Zhou et al., 2025; Wang et al., 2025). *Decoding* strategies encourage self-correction and uncertainty expression, including self-consistency sampling (Wang et al.,

*Table 13.* Comparison of TruthRL with related methods. **Training** indicates the optimization paradigm used: SFT-based methods use static supervision, while RL-based methods (PPO, GRPO) learn from online reward signals. **Reward Type** describes the training signal used to guide learning. In the reward formulas, $\mathbb{I}_{correct}$ denotes the binary correctness indicator and $c \in [0,1]$ denotes the model's confidence score. **Uncertainty Expression** describes how the model communicates uncertainty: calibration-based methods output a numerical confidence score alongside the answer (*i.e.*, the model always answers), whereas TruthRL generates a verbalized natural language response such as "I don't know" (*i.e.*, the model may choose not to answer at all). **Over-Abstention Control** describes the mechanism that prevents the model from abstaining excessively: TruthRL's group-relative advantage dynamically adjusts the credit assigned to abstention based on the composition of sampled responses within each GRPO group. Given the above differences, these methods are orthogonal to our work: in addition to the pursuit of accuracy, they improve *how confidently* the model answers, whereas TruthRL improves *whether* the model answers at all. This distinction is important because even a perfectly calibrated model that always answers will still produce hallucinations on questions beyond its knowledge, while empowering the model to abstain can avoid any potential hallucinations in such cases.

| Method | Training | Reward Type | Uncertainty Expres. | Over-Abst. Control |
|---|---|---|---|---|
| RLCR | GRPO | Binary-based Scheme $r = \mathbb{I}_{correct} - (c - \mathbb{I}_{correct})^2$ | Confidence Score | N/A (always answer) |
| SaySelf | SFT + PPO | Binary-based Scheme $R = 1 - 2(\mathbb{I}_{correct} - c)^2$ | Confidence Score | N/A (always answer) |
| RewardingDoubt | PPO | Binary-based Scheme $r = \mathbb{I}_{correct} \cdot \log c + (1 - \mathbb{I}_{correct}) \cdot \log(1-c)$ | Confidence Score | N/A (always answer) |
| **TruthRL (Ours)** | **GRPO** | **Ternary-based Scheme** $r \in \{-1, 0, +1\}$ | **Natural Language** | **Dynamic Credit** |

2023), calibrated decoding (Kadavath et al., 2022; Whitehead et al., 2022), and contrastive decoding (Chuang et al., 2024). *Fine-tuning* approaches seek to instill more truthful behavior directly into the model's parameters (Tian et al., 2024; Li et al., 2025c; Zheng et al., 2025; Li et al., 2025a; Newman et al., 2025). Common methods include SFT (Zhou et al., 2023), DPO (Tian et al., 2024; Zhang et al., 2024b; Xu et al., 2025), and reinforcement learning from human feedback (RLHF) (Ouyang et al., 2022) on curated datasets of high-quality, factual question-answer pairs. While these methods can enhance accuracy on in-distribution topics, their generalization might degrade significantly on out-of-distribution questions (Kirk et al., 2024). Our work addresses a key limitation of many existing approaches: they do not explicitly train models to recognize *when* to abstain. Among prior efforts, the most closely related is R-Tuning (Zhang et al., 2024a), which likewise aims to reduce hallucinations. However, as we show in Section 4, its reduction of hallucination comes at the cost of substantially reduced coverage. This trade-off underscores the need for training frameworks that directly optimize for truthfulness—striking a balance between factual accuracy and calibrated abstention, thereby minimizing the risk of misleading outputs.

**Confidence Calibration vs. Abstention** A complementary line of work focuses on improving LLMs' confidence calibration, which trains models to express well-calibrated uncertainty estimates alongside their answers, rather than teaching them to abstain outright (Kapoor et al., 2024). RLCR (Damani et al., 2026) augments binary correctness rewards with a Brier score term, training the model to output numerical confidence values that align with its actual accuracy. While effective for calibration, RLCR does not teach the model to *refuse to answer* (Xin et al., 2021; Cohen et al., 2024). Instead, it always produces a response and merely adjusts the accompanying confidence score. SaySelf (Xu et al., 2024b) trains models to generate self-reflective rationales about their own confidence before answering, using SFT on model-generated reasoning chains. The model learns to verbalize its uncertainty level but still always provides an answer. In other words, the calibration signal is in the expressed confidence, not in the decision to abstain. RewardingDoubt (Bani-Harouni et al., 2026) uses RL to reward models for producing calibrated confidence expressions, optimizing the alignment between stated confidence and actual correctness. Again, the model always answers because the training signal rewards accurate self-assessment rather than selective silence. Therefore, these approaches are complementary to our work: they improve *how confidently* the model answers, whereas TruthRL improves *whether* the model answers at all. TruthRL's ternary reward structure ($+1$ for correct, $0$ for abstain, $-1$ for hallucination) explicitly incentivizes the model to generate "I don't know" responses when uncertain, and GRPO's group-relative advantage mechanism dynamically calibrates when abstention is beneficial. This distinction is important because even a perfectly calibrated model that always answers will still produce hallucinations on questions beyond its knowledge. A detailed comparison is provided in Table 13.

**Reinforcement Learning for LLMs** Reinforcement learning (RL) has become a central paradigm for post-training LLMs, enabling alignment beyond supervised fine-tuning. The most prominent example, RLHF (Ouyang et al., 2022; Christiano et al., 2017; Rafailov et al., 2023), encodes user preferences into reward models and has produced systems that are generally

more helpful, safer, and better aligned. More recently, Reinforcement learning from verifiable rewards (RLVR) (Shao et al., 2024; Zhu et al., 2025; Shao et al., 2025; Lambert et al., 2024; Guo et al., 2025; Wei et al., 2026) has shown that binary reward signals (correct vs. incorrect) can elicit sophisticated chain-of-thought reasoning. However, this formulation conflates abstention with error, thereby discouraging models from producing calibrated "I don't know" responses (Song et al., 2025). To alleviate such limitations, several extensions introduce richer reward structures, including uncertainty-aware RL (Xu et al., 2024a; Xue et al., 2024; Lin et al., 2024; Wang et al., 2024c; Li et al., 2025b; Liang et al., 2024) and multi-objective optimization for factual faithfulness (Wang et al., 2024a; Chen et al., 2025b). Despite these advances, designing scalable reward signals that reliably capture truthfulness while balancing accuracy and uncertainty remains an open challenge. In this work, we demonstrate that reward structure—binary vs. ternary, whether and how uncertainty is incorporated—can fundamentally influence whether models learn to balance factual accuracy with abstention.

## C. Implementation Details

**Experimental setup.** We follow the retrieval setup from CRAG (Yang et al., 2024a), using up to 50 web pages as retrieval documents per question. For each question, the question text is used as the search query, and up to 50 HTML pages are stored from the search API. For NaturalQuestions (NQ), HotpotQA, and MuSiQue, we use the 2018 Wikipedia dump (Karpukhin et al., 2020) as the knowledge source and employ the E5 retriever (Wang et al., 2024b), as in line with the Search-R1 setup (Jin et al., 2025). Following prior works (Yang et al., 2024a; Huang et al., 2025; Kachuee et al., 2025), we set $w_1 = 1, w_2 = 0, w_3 = 1$ for truthfulness score calculation (defined in Section 2.1). The correctness of predicted answers are judged by Llama3.3-70B-Instruct against the reference answers.

**DPO.** DPO (Rafailov et al., 2023) is an offline RL method that trains the model to prefer certain responses over others. Preference pairs are constructed differently for OOK and non-OOK questions. For OOK questions, the preferred response is "I don't know," and the dispreferred response is a randomly chosen incorrect answer. For non-OOK questions, correct and incorrect responses are paired. DPO expresses the probability of preference data with the policy model rather than the reward model, yielding the following objective:

$$\mathcal{L}_{\text{DPO}}(\theta) = -\mathbb{E}_{(x, y_w, y_l) \sim \mathcal{D}} \left[ \log \sigma \left( \beta \log \frac{\pi_\theta(y_w \mid x)}{\pi_{\text{ref}}(y_w \mid x)} - \beta \log \frac{\pi_\theta(y_l \mid x)}{\pi_{\text{ref}}(y_l \mid x)} \right) \right],$$

where $(x, y_w, y_l)$ are preference pairs consisting of the prompt, the winning response, and the losing response from the preference dataset $\mathcal{D}$.

**Iterative DPO.** This variant builds on a DPO-trained checkpoint and iteratively constructs preference pairs in the same way over the training set.

**Training details.** Our models are trained on 8 NVIDIA H100 GPUs with 80GB memory using full-parameter fine-tuning. By default, we use the Open-R1 library (Hugging Face, 2025) as the training framework. To optimize GPU utilization, we adopt DeepSpeed (Rajbhandari et al., 2020) with ZeRO-3 offload, along with gradient checkpointing, FlashAttention-2 (Dao, 2024), and bf16 mixed-precision training enabled. To optimize model performance, we conduct an extensive hyperparameter search with batch sizes in [16, 32, 64], learning rates in [5e-7, 1e-6, 2e-6, 3e-6, 5e-6, 1e-5], and training epochs in [1, 2, 3].

For SFT, RFT, and R-Tuning, we use a learning rate of 5e-6 and a batch size of 16, with a cosine learning rate scheduler and 3% warmup steps, trained for 1 epoch. For DPO and iterative DPO, we use a learning rate of 3e-6 and a batch size of 32, trained for 1 epoch.

For RL training, we use the VeRL framework (Sheng et al., 2024) with a constant learning rate of 1e-6, and a batch size of 64. The KL divergence regularization coefficient $\beta$ and clip ratio $\epsilon$ are set to 0.001 and 0.2, respectively. The maximum context length and number of generated tokens are set to 16,384 and 2,048. For efficient LLM rollouts, we use vLLM (Kwon et al., 2023) with a tensor parallel size of 2 and a GPU memory utilization ratio of 0.8. Rollout sampling is performed with temperature = 1.0 and top-p = 1.0. The maximum token length for all models is fixed at 16k. We set $\lambda = 0.5$ in Section A.

**Inference details.** We use vLLM for efficient inference and adopt greedy decoding (*i.e.*, temp = 0) for evaluation to ensure reproducible results. For data construction in RFT, we sample 64 responses with a temperature of 0.6 and top-p of 0.9. The maximum token length at inference is set to 32k.

## D. Prompt Template

**Inference prompts**. Below we present the inference prompts for both without and with retrieval setups in Table 14 and Table 15, respectively.

*Table 14.* Inference prompt under without retrieval setup.

---

**Inference Prompt (Without Retrieval)**

**Input:** You are given a Question and the time when it was asked in the Pacific Time Zone (PT), referred to as "Query Time". The query time is formatted as "mm/dd/yyyy, hh:mm:ss PT". Your task is to answer the question based on factual information in your own knowledge.

Please adhere to the following guidelines when formulating the answer:
1. If the question contains a false premise or assumption, answer "invalid question".
2. If you are uncertain or don't know the answer, answer "I don't know".

Please reason step by step and then provide the final answer. The reasoning process must be enclosed within <think> </think> tags. The final answer MUST be put in \boxed{}. For example, \boxed{I don't know}, \boxed{invalid question}, \boxed{3 times}, \boxed{New York}, etc.

### Question: {question}
### Query Time: {query time}

**Output:** {answer}

---

*Table 15.* Inference prompt under with retrieval setup.

---

**Inference Prompt (With Retrieval)**

**Input:** You are given a Question, References and the time when it was asked in the Pacific Time Zone (PT), referred to as "Query Time". The query time is formatted as "mm/dd/yyyy, hh:mm:ss PT". The references may or may not help answer the question. Your task is to answer the question based on factual information in the references or your own knowledge.

Please adhere to the following guidelines when formulating the answer:
1. If the question contains a false premise or assumption, answer "invalid question".
2. If you are uncertain or don't know the answer, answer "I don't know".

Please reason step by step and then provide the final answer. The reasoning process must be enclosed within <think> </think> tags. The final answer MUST be put in \boxed{}. For example, \boxed{I don't know}, \boxed{invalid question}, \boxed{3 times}, \boxed{New York}, etc.

### Question: {question}
### Query Time: {query time}
### References: {documents}

**Output:** {answer}

---

**LLM-as-a-judge prompts**. Below we present the judge prompts for outcome and reasoning quality in Table 16 and Table 17, respectively.

## E. Case Study

Table 18 presents a representative example illustrating the accuracy of the LLM-based judge.

*Table 16.* LLM-as-a-judge prompt for evaluating outcome.

| LLM-as-a-judge Prompt (Outcome) |
| --- |
| **Input:** Assume you are a human expert in grading predictions given by a model. You are given a question and a model prediction. Judge if the prediction matches the ground truth answer by following these steps:
1: Take it as granted that the Ground Truth is always correct.
2: If the Prediction exactly matches the Ground Truth, "score" is 1.
3: If the Prediction does not exactly match the Ground Truth, go through the following steps and likely give a score as 0.
4: If the Ground Truth is a number, "score" is 1 if and only if the Prediction gives a number that almost exactly matches the ground truth.
5: If the Prediction is self-contradictory, "score" must be 0.
6: If the prediction is not answering the question, "score" must be 0.
7: If the prediction is a concise and correct summary of the ground truth, "score" is 1.
8: If ground truth contains a set of items, prediction must contain exactly same items for the score to be 1.
9: Otherwise, "score" is 0.

Output a JSON blob with an "explanation" field explaining your answer as short as possible and an "score" field with value 1 or 0.

You should make the judgment based on provided examples.

### Examples: {examples}
### Question: {question}
### Ground Truth: {ground truth}
### Prediction: {predicted answer}

**Output:** {judgment} |

*Table 17.* LLM-as-a-judge prompt for evaluating reasoning quality.

| LLM-as-a-judge Prompt (Reasoning Quality) |
| --- |
| **Input:** Assume you are a human expert in evaluating the usefulness of model-generated reasoning. You are given a question and a model-generated reasoning. Judge if the reasoning provides precise information to correctly answer the question by following these steps:
1: Evaluate if the reasoning directly addresses the question.
2: Check if the key points in the reasoning are relevant to the query.
3: If the reasoning provides precise and relevant information, "score" is 1.
4: If the reasoning is vague, unrelated, or does not address the question, "score" is 0.

Output a JSON blob with an "explanation" field explaining your answer as short as possible and an "score" field with value 1 or 0.

You should make the judgment based on provided examples.

### Examples: {examples}
### Question: {question}
### Ground Truth: {ground truth}
### Reasoning: {predicted reasoning}

**Output:** {judgment} |

*Table 18.* An example of LLM-as-a-judge result.

---

**LLM-as-a-judge Result**

**Question:** How many times has Lizzo won the Grammy Award for Album of the Year?

**Ground Truth:** Lizzo has never won the Grammy Award for Album of the Year, though she has been nominated twice.

**Prediction:** <think> To answer the question "How many times has Lizzo won the Grammy award for album of the year?", we need to analyze the information provided in the references. Document [4] mentions that Lizzo has won 4 Grammy Awards, but it does not specifically mention the category of Album of the Year. However, Document [5] mentions that Lizzo's album "Special" is nominated for Album of the Year, but it does not mention that she has won the award. After reviewing all the documents, I found that none of them explicitly state that Lizzo has won the Grammy Award for Album of the Year. Therefore, the answer is 0, as there is no evidence to suggest that Lizzo has won the Grammy Award for Album of the Year. </think> \boxed{0}.

**LLM judge result:** {"score": 1, "explanation": "The prediction correctly states the number of times Lizzo has won the Grammy Award for Album of the Year, which is zero."}

---

