# OpenReview forum: "TruthRL: Incentivizing Truthful LLMs via Reinforcement Learning"
_ICML.cc/2026/Conference — ICML 2026 regular_

### Official Review · Reviewer_FY1B · 2026-02-27

**Soundness:** 3
**Presentation:** 3
**Significance:** 2
**Originality:** 2
**Overall Recommendation:** 4
**Confidence:** 4

**Summary:**

This paper proposes a ternary-reward RL framework called TruthRL that explicitly optimizes truthfulness by rewarding correct answers, penalizing hallucinations, and permitting abstention. The core idea is simple and empirically effective, showing meaningful hallucination reduction across several QA benchmarks. Advancing prior work that explores abstention training, e.g., R-Tuning, IDK-token modeling, TruthRL primarily contributes a reward-shaping instantiation within RL.

**Compliance With Llm Reviewing Policy:**

Affirmed.

**Final Justification:**

During rebuttal, I reiterated two critical issues that the paper has missed in their initial response:

- Non-QA experiment is still limited. It remains unclear if the method generalizes to broader and fact-sensitive generation tasks, such as fact verification.
- Human evaluations are required to ensure authentic gains.

The authors then conducted further experiments on fact verification based on FEVER dataset. Their results indicate that TruthRL generalizes well to fact verification beyond QA and Math reasoning tasks. Furthermore, they conducted a human evaluation on the CRAG test set with 643 questions and reassessed answers generated by Prompting and TruthRL using Llama3.1-8B-Instruct. Result confirms the gains are real according to human judge.

Therefore, I raised my overall recommendation from 3 to 4.

**Key Questions For Authors:**

- Apart from the specific reward shaping, how does it fundamentally differ from prior abstention-aware or selective prediction training methods such as R-Tuning and IDK-token training? What conceptual advance is introduced?
- Does it improve truthfulness beyond extractive QA in the context of long-form generation, summarization, and multi-step reasoning? If not tested, the general claims should be tempered?
- How does it compare against a calibrated selective prediction baseline trained with confidence thresholds rather than RL?
- Are results stable across different correctness judges or human evaluations?
- How does it control excessive abstention?

**Limitations:**

No. The paper does not discuss limitations or potential negative societal impacts. The authors are suggested to:

- Discuss the possibility that models may become overly conservative, reducing usefulness in high-stakes domains.
- Address the risk that optimizing abstention via RL could lead to reward hacking, where models may avoid difficult questions.
- Acknowledge that improvements rely heavily on how hallucination and correctness are judged, and that metric bias could shape behavior.
- Clarify whether gains generalize beyond QA benchmarks to open-ended or multilingual settings.

**Strengths And Weaknesses:**

Strengths
- The ternary reward under GRPO is easy to understand and appears to materially reduce hallucinations while keeping accuracy competitive across multiple QA benchmarks including CRAG, NQ, HotpotQA, and MuSiQue.
- The paper is clearly motived by highlighting how SFT / RL can suppress abstention and promote confident guessing, and supports this with behavioral breakdowns with binary vs ternary reward.
- Reasonable analyses around abstention behavior by comparing with knowledge boundary probing baselines and studying abstention dynamics and robustness on hallucination-baiting questions.

Weaknesses
- Novelty is incremental relative to existing literatures, such as abstain to reduce hallucination and teach IDK. Prior work already targets abstention/unknown recognition. For instance, R-Tuning teaches "I don’t know" behavior [1]; AbstainQA-style work studies knowledge-gap detection and abstention [2]; [IDK] token work explicitly models uncertainty [3]. The difference of TrustRL is mainly reward shaping in RL (ternary), which is not groundbreaking.

- Connection to established selective prediction / reject option is under-positioned. The paper’s abstention framing aligns with selective prediction in NLP [4], but it doesn’t clearly situate TruthRL as an RL instantiation of this line or compare to strong selective-prediction style training/evaluation protocols.

- Evaluation scope may overclaim general truthfulness. Results are QA-centric, and it remains unclear whether the method transfers to other truthfulness-sensitive generation tasks such as summarization, long-form, and tool-use, or whether it mainly learns "abstain more safely" under the specific judge/metric setup.

- Improvements rely on how correctness/hallucination / abstention are judged and implemented. Without strong human evaluation or multiple independent verifiers across settings, it’s hard to ensure gains are not partly metric- / judge-specific.

[1] R-Tuning: Instructing Large Language Models to Say ‘I Don’t Know’. NAACL 2024

[2] Don't Hallucinate, Abstain: Identifying LLM Knowledge Gaps via Multi-LLM Collaboration. ACL 2024

[3] I Don’t Know: Explicit Modeling of Uncertainty with an [IDK] Token. NeurIPS 2024

[4] The Art of Abstention: Selective Prediction and Error Regularization for Natural Language Processing. ACL-IJCNLP 2021

---

> ### Author Rebuttal · Authors · 2026-03-31
>
> We sincerely thank the reviewer for the constructive feedback.
>
> **[Q1]: How does TruthRL fundamentally differ from prior abstention-aware or selective prediction training methods such as R-Tuning and IDK-token training? What conceptual advance is introduced?**.
>
> **[A1]**: We appreciate the reviewer’s suggestion on highlighting the differences between TruthRL and prior works. We provide a comparison table (Table 1) in this [rebuttal PDF](https://anonymous.4open.science/r/TruthRL-ICML2026-Submission/To_Reviewer_FY1B.pdf). The key differences include:
>
> - AbstainQA operates at inference time via multi-LLM collaboration without modifying weights. TruthRL internalizes abstention through training, requiring only a single inference pass
>
> - R-Tuning pre-identifies unknown questions and creates static SFT labels, which are never updated during training. TruthRL continuously discovers knowledge boundaries through online RL exploration.
> - IDK-Token is a continued pretraining method that modifies the model vocabulary and requires a full pretraining on a large corpus with up to 177M training samples. TruthRL is a post-training RL method with no architectural changes, using only 656 samples.
>
> We have cited these papers and added discussions in the paper revision.
>
> - **The unique conceptual advance of TruthRL is the dynamic credit assignment for abstention**: TruthRL combines ternary rewards with GRPO's group-relative advantage, so the same abstention can receive different credit depending on group composition (as exemplified in **Section 3.2**). As also shown in **Figure 3**, the binary scheme suppresses abstention to near-zero, while the ternary scheme produces adaptive abstention dynamics in both pro- and anti-abstention groups, exhibiting behaviors that cannot be reproduced by SFT-based methods, inference-time approaches, or RL with binary rewards.
>
>
> **[Q2]: Insufficient discussion of selective prediction [4]. How does truthRL compare against a calibrated selective prediction baseline?**
>
> **[A2]**:  We have added a discussion of selective prediction [4] in our revision. In this line, predictions below a confidence threshold are rejected as abstentions, but the model itself is never trained to abstain. **Under this context, TruthRL can be viewed as learning the optimal selective prediction policy through RL**, with two key distinctions:
> - Abstention is internalized within the model's generation process rather than applied by an external filter
> - No confidence threshold is needed, avoiding reliance on potentially miscalibrated confidence estimates
>
> As suggested, we compare TruthRL against a selective prediction baseline under different thresholds in Table 2 of the same rebuttal PDF.
>
>
> **[Q3]: Results are QA-centric. Does it improve truthfulness beyond extractive QA?**
>
> **[A3]**: We’d like to note that our evaluation spans diverse formats beyond extractive QA: single-hop factoid (NQ), multi-hop reasoning (HotpotQA), multi-step compositional (MuSiQue), and web-based QA (CRAG).
>
> Per the reviewer's suggestion, we further validate on a non-QA benchmark GSM8K (mathematical reasoning) in Table 3 of the same rebuttal PDF.
>
>
> **[Q4]: Are results stable across different correctness judges or human evaluations?**
>
> **[A4]**:  We clarify that our LLM-judge setup exactly follows the CRAG benchmark (>99% human agreement). **Table 5** further evaluates with three independent judge families (Llama-3.3-70B, Qwen2.5-72B, Gemma-3-27B), where TruthRL consistently achieves the best truthfulness and lowest hallucination, showing stable performance across different judges.
>
>
>
> **[Q5]: How does TruthRL control excessive abstention?**
>
> **[A5]**:  As detailed in **Section 3.2**, TruthRL achieves this control through GRPO's group-relative advantage estimate: when correct responses exist in the group, abstention receives negative advantage, discouraging unnecessary abstention. **Figure 5(b)** verifies this in action: early in training, few correct responses are generated, so abstention is encouraged and rises quickly. As the model evolves and generates more correct answers, abstention is accordingly penalized, preventing excessive conservatism.
>
> **[Q6]: Suggestions on discussing limitations or potential negative societal impacts.**
>
> **[A6]**: Thanks for the thoughtful suggestions. ​​Some of these discussions are scattered across our paper, and we have now added a new section in the revision to demonstrate how we could address the potential limitations: (1) over-conservative behavior (see Table 7), (2) reward hacking (see Table 5), (3) judge dependence and metric bias (see Tables 4 & 6), and (4) generalization beyond QA (see additional GSM8K results in rebuttal).
>
>
> Please let us know if you have any further questions, and we are happy to incorporate additional suggestions you might have. If you find our response satisfactory, we would be grateful if you could consider raising your score. Thanks again for your time and efforts!

---

> > ### Author Rebuttal · Reviewer_FY1B · 2026-04-01
> >
> > 1. Non-QA experiment is still limited. It remains unclear if the method generalizes to broader and fact-sensitive generation tasks, such as fact verification.
> > 2. Human evaluations are required to ensure authentic gains.

---

> > > ### Author Response · Authors · 2026-04-03
> > >
> > > Thanks for acknowledging our response. We address both remaining concerns below.
> > >
> > > - **Beyond QA and Math, TruthRL also generalizes well to fact verification.** We previously showed that TruthRL can generalize to non-QA tasks such as math reasoning. In this round, we further validate TruthRL on a widely used fact verification task FEVER [1]. It requires the model to verify a given claim with retrieved evidence as input, and determine whether the claim is supported (SUP), refuted (REF), or not enough information (NEI). We evaluate TruthRL and baseline methods on the full test set (9,999 samples, 3,333 per class) of FEVER and report accuracy under each category. For fair comparison, we use Llama-3.1-8B-Instruct as the backbone across all compared methods.
> > >
> > > | Model | Overall | SUP | REF | NEI |
> > > |----------|:-------:|:----:|:----:|:----:|
> > > | Prompting | 88.2% | 91.9% | 72.9% | 99.9% |
> > > | SFT | 90.2% | 96.2% | 79.9% | 94.4% |
> > > | RFT | 87.6% | 93.8% | 78.7% | 90.1% |
> > > | R-Tuning | 90.4% | 96.0% | 80.3% | 94.9% |
> > > | TruthRL | **92.6%** | 93.5% | 87.4% | 96.9% |
> > >
> > > TruthRL achieves the best overall accuracy (92.6%), validating the generalization of our method to fact verification. Notably, TruthRL achieves the highest REFUTED accuracy (87.4% vs. ≤80.3% for all baselines). **This suggests TruthRL excels at identifying contradictions between claims and evidence, which is a core capability for fact-sensitive tasks**. Interestingly, Prompting achieves near-perfect NEI accuracy because NEI claims are presented without any evidence (since no supporting or refuting evidence exists by definition), making the absence of evidence a trivial signal for the model to recognize insufficient information. Fine-tuned models show degraded NEI accuracy, suggesting that fine-tuning tends to encourage over-confident predictions even in the absence of evidence. Compared to other fine-tuned baselines, TruthRL is more effective in mitigating such tendency, maintaining competitive NEI accuracy.
> > >
> > > - **LLM-based judge aligns well with human evaluation**. To ensure authentic gains, we conducted a human evaluation on the CRAG test set (643 questions), re-judging answers generated by Prompting and TruthRL with Llama3.1-8B-Instruct as the backbone.
> > >
> > > | Judge | Truthfulness (↑) | Hallucination (↓) | Uncertain (-) | Accuracy (↑) |
> > > |--------|:----------------:|:------------------:|:-------------:|:------------:|
> > > |        |                  | **Prompting**      |               |              |
> > > | LLM    | 5.3             | 43.5               | 7.7           | 48.8         |
> > > | Human  | 3.1             | 44.6               | 7.7           | 47.7         |
> > > |        |                  | **TruthRL**        |               |              |
> > > | LLM    | 37.2            | 19.4               | 24.0          | 56.6         |
> > > | Human  | 36.9            | 19.4               | 24.3          | 56.3         |
> > >
> > > For Prompting, the human-LLM agreement is 636/643 = 98.9%. For TruthRL, the human-LLM agreement is 641/643 = 99.7%, **confirming the finding in CRAG [2] that the LLM-based judge and human evaluation are highly aligned.**
> > >
> > > Though mismatched cases are rare, we further conducted an error analysis on such cases and identified two main LLM judge error patterns:
> > >
> > > - Ambiguity in acceptable answers: The LLM judge accepted answers that are related but not equivalent to the reference.
> > > > **Question**: Can you tell me the name of the artist who won the award for Song of the Year in the 49th Grammy (2006)? \
> > > > **Reference answer**: Dixie Chicks \
> > > > **Model prediction**: Natalie Maines \
> > > > **LLM judge**: {score: 1, explanation: Natalie Maines is the lead singer of the Dixie Chicks.} \
> > > > **Human judge**: {score: -1, explanation: The award was given to Dixie Chicks as a group, not to Natalie Maines individually.}
> > > - Over-leniency on numerical precision: The LLM judge accepted predictions with different numerical values by claiming they "match after rounding."
> > > > **Question**: What percentage of gross for Pixar's 2020 movies came from outside the US and Canada? \
> > > > **Reference answer**: 74.57% \
> > > > **Model prediction**: 74% \
> > > > **LLM judge**: {score: 1, explanation: The prediction exactly matches the ground truth after rounding.} \
> > > > **Human judge**: {score: -1, explanation: 74% and 74.57% are different values.}
> > >
> > > The error analysis demonstrates that these rare mismatches are largely due to the inherent ambiguity in open-ended QA evaluation rather than systematic bias associated with the LLM judge, and the overall >98% human-LLM agreement confirms that **the LLM-based judge provides a reliable and scalable evaluation of model truthfulness.**
> > >
> > > If you find our response satisfactory, we would be grateful if you could consider raising your score. Thanks again for your time!
> > >
> > >
> > > **References**
> > >
> > >
> > > [1]. Thorne et al. FEVER: a Large-scale Dataset for Fact Extraction and VERification. NAACL 2018.
> > >
> > > [2]. Yang et al. CRAG -- Comprehensive RAG Benchmark.  NeurIPS 2024.

---

### Official Review · Reviewer_pwwY · 2026-03-11

**Soundness:** 3
**Presentation:** 2
**Significance:** 4
**Originality:** 3
**Overall Recommendation:** 5
**Confidence:** 3

**Summary:**

The paper addresses the problem of hallucinations in Large Language Models (LLMs). The authors claim that supervised fine-tuning or traditional Reinforcement Learning (RL) approaches do not generalise well and reinforce hallucinations by incentivsing accuracy over everything else.
Their method, *TruthRL*, leverages RL to fine-tune LLMs for hallucination reduction, introducing "a truthfulness-driven ternary reward design that explicitly rewards correct answers, penalises hallucinations, and treats abstentions as neutral."
The authors perform extensive empirical evaluation, showing that *TruthRL* with this ternary reward...
 1. ...reduces **hallucinations** compared to *zero-shot prompting*-, *supervised fine-tuning*-, and *RL*-based baselines, as well as *TruthRL* with a binary reward (section 4.2).
 2. ...improves **truthfulness** as measured by a weighted sum of accuracy, hallucination rate, and uncertainty rate (section 4.2).

Furthermore, they claim that their method abstains not because of "conservativism," but genuinely recognises the LLMs knowledge boundary and support this claim with small experiments (section 4.3).
Finally, they show ablation results, justifying their choice of reward function.

**Compliance With Llm Reviewing Policy:**

Affirmed.

**Final Justification:**

The authors addressed all my comments, so I retain my score of "accept".

**Key Questions For Authors:**

1. Which models did you benchmark in sections 2.2 and 2.3? Are these the state of the art?
2. You introduce the weights of the truthfulness score as "$w_1, w_2, w_3 \ge 0$," (page 3), but then ablate "$w_2 \in \\{-.5, 0, .5\\}$." Can you motivate this choice of ablation, and why negative weighting for the *uncertainty rate* makes sense?

**Limitations:**

yes

**Strengths And Weaknesses:**

## Strengths:
- **Significance:** The paper addresses the highly important issue of LLM hallucinations and presents a convincing strategy towards improving the problem.

- **Originality:** To the best of my understanding, the novelty of the paper mainly results from the proposed reward function for RL-based fine-tuning of LLMs. Given the promising results, this is sufficient in my opinion.

- **Soundness:** In general, the method is sound, and the evaluation seems solid. The authors also provide a link to their code. There are, however, some vague parts that should be clarified (see below).

## Weaknesses:
- **Presentation:**
  While the paper has a nice flow and is well readable, there are multiple parts that are vague or leave out details. Specific comments on presentation below:
  - The abstract is too long: 11 instead of "4–6 sentences" as requested in the [template](https://media.icml.cc/Conferences/ICML2026/Styles/example_paper.pdf). This needs to be addressed for the final submission.
  - The reporting of baseline results in sections 2.2 and 2.3 is in an unconventional part of the paper. This is OK, since it fits the argumentation, but the authors should provide a short introduction (1-2 sentences) in the beginning of the section to explain why results are reported here.
  - In general, the content of sections 2.2 and 2.3 needs more details. Majority@k needs to be introduced, and the specific tested methods need to be referenced.
  - More details are needed in the beginning of Section 3.1, where the authors write that they "sample 256 responses, and the question is marked as OOK if none of the responses is correct." It needs to be clear how sampling is done. Also report the values of $temperature$ and $p$ used for sampling.
  - Optional improvements:
    - In Table 1, the best *hallucination rates* and *accuracy* results should also be bold to improve readability.
    - The formulation of the second contribution (on page 2) is redundant. Some rephrasing can improve this.

- **Soundness:**
  While the approach is generally sound, the presentational problems of the paper sometimes extend to the soundness domain. I expect, however, that these can be easily fixed by clarifying things in the text:
  - Metrics used must be more thoroughly introduced:
  **(i)** The weights chosen for the weighted sum truthfulness score need to be explicitly named *early on* in the paper (especially, since the authors seem to set the weight $w_2$ = 0), and
  **(ii)** the *majority@k* metric must be introduced as it is not a standard evaluation metric.
  - Baselines for section 2.3 need to be named before or in this section (see above).
  - There should a statement why sampling is chosen over deterministic generation (i.e. $temperature$=0 and $p$=0).
  - Sampling specifics for OOK questions need to be specified (see above).

---

> ### Author Rebuttal · Authors · 2026-03-31
>
> We thank the reviewer for the positive assessment, the recognition of the paper's significance and sufficient novelty, and the very detailed feedback. We have incorporated all suggested improvements.
>
> **[Q1]: Suggestions on improving presentation.**
>
> **[A1]**: We sincerely appreciate the reviewer's careful review of our paper. Per the reviewer's suggestions, we've made the following updates and clarifications in the revision.
>
> - Substantially shortened the abstract to improve conciseness.
> - Expanded the discussion of Figure 2 and Figure 3, explaining why they are reported in Sections 2.2 and 2.3.
> - Added the introduction of Majority@k and explicit references to the tested methods in Sections 2.2 and 2.3.
> - Added more details at the beginning of Section 3.1 to clarify the sampling setup, including the values of temperature and top_p.
> - Updated Table 1 to highlight the best hallucination rates and accuracy.
> - Rephrased the second contribution on Page 2 to avoid redundant statements.
>
> **[Q2]: Suggestions on improving soundness.**
>
> **[A2]**: Following our response to Q1, we've added the following clarifications to make the implementation details clearer:
>
>
> - The default weights for the truthfulness score formula are now introduced early in the Preliminary section (Section 2). We also introduced the Majority@k metric before it is first reported.
> - All baselines in Section 2.3 are now explicitly named with references.
> - We clarified that sampling is only used to identify out-of-knowledge (OOK) questions as a preliminary step for methods that rely on it. For the main comparison, we use greedy decoding, as stated in Appendix C (Implementation Details). We've added sampling specifics (e.g., temperature and top_p) for OOK identification in Section 3.1 and added a statement to make this distinction clear.
>
> **[Q3]: Which models benchmarked in sections 2.2 and 2.3? Are these state of the art?**
>
> **[A3]**: As stated in the caption of Figure 2, we use Llama-3.1-8B-Instruct, one of the state-of-the-art models at its scale, as the primary backbone for the studies in Sections 2.2 and 2.3. We've added clarifications in these sections and updated the captions of both Figure 2 and Figure 3 to make this clearer.
>
>
> **[Q4]: Weights introduced as $w_₁, w_₂, w_₃ ≥ 0$, but ablation uses $w_₂ \in $ {−0.5, 0, 0.5}. Motivate this choice and why negative weighting for uncertainty rate makes sense?**
>
> **[A4]**: Thank you for bringing this up. Our original notation defines $w_₁, w_₂, w_₃ ≥ 0$, assuming that abstention is not explicitly penalized in most deployments. However, the treatment of abstention may vary depending on application scenarios, which motivates our ablation study.
>
>
> In our revised paper, we've updated the notation in Section 3.1 to clarify that $w_₁, w_₃ ≥ 0$ (accuracy is always rewarded; hallucination is always penalized), while $w_₂$ is determined by deployment needs: $w_₂ > 0$ rewards abstention (safety-critical settings), $w_₂ = 0$ treats it as neutral (our default, following standard practice), and $w_₂ < 0$ penalizes abstention (high-coverage settings). Our ablations in **Table 4** and **Table 6** show that TruthRL remains strong across all three settings, demonstrating robustness to different preferences. We've also added the default values for these weights early on in the paper for better clarity.
>
>
> We hope the reviewer finds our response helpful. We are also happy to incorporate additional suggestions you might have. Thanks again for your time and the constructive feedback!

---

> > ### Author Rebuttal · Reviewer_pwwY · 2026-04-02
> >
> > Thank you for your response. All my points have been adequately addressed.

---

> > > ### Author Response · Authors · 2026-04-03
> > >
> > > We are glad that our response addressed the reviewer’s concerns! Thank you again for the support and constructive comments. We will ensure to include the suggestions in our final revision.

---

### Official Review · Reviewer_3oXR · 2026-03-12

**Soundness:** 2
**Presentation:** 2
**Significance:** 2
**Originality:** 2
**Overall Recommendation:** 4
**Confidence:** 3

**Summary:**

This paper tackles a real and timely problem: making LLMs more truthful by enabling them to abstain when uncertain rather than hallucinate. The authors propose TruthRL, a reinforcement learning framework with a ternary reward scheme that treats correct answers (positive), hallucinations (negative), and abstentions (neutral) differently. The intuition is straightforward, reward correct responses, penalize guessing incorrectly, and neither reward nor penalize refusals. Experiments on CRAG, NaturalQuestions, HotpotQA, and MuSiQue using Llama3.1-8B and Qwen2.5-7B show reduced hallucination rates and improved truthfulness scores compared to baselines.

**Compliance With Llm Reviewing Policy:**

Affirmed.

**Final Justification:**

Raised score from 3 to 4.

**Key Questions For Authors:**

Please see weaknesses.

**Limitations:**

Please see weaknesses.

**Strengths And Weaknesses:**

**Strengths**

* Hallucination remains a key barrier to LLM deployment, and the paper rightly emphasizes that truthfulness involves knowing when to say "I don't know." This framing is practically relevant and timely.

* The ternary scheme is simple but effective. Explicitly distinguishing hallucinations from abstentions gives the model a clearer signal than standard binary reward approaches.

* The evaluation covers multiple datasets, both retrieval and non-retrieval settings, and includes ablation studies. The consistent reduction in hallucination rates is convincing.

**Weaknesses**

* Truthfulness metric uses manually tuned weights. The composite score weights accuracy, abstention, and hallucination rates based on author choices. Different applications might prefer different trade-offs, so it's unclear how well the results generalize beyond this specific weighting.

* TruthRL modifies the reward function but relies on existing RL frameworks. It's not entirely clear how much of the gain comes from the ternary reward versus other implementation details.

* Treating abstention as always neutral assumes this is the right default. But some applications may prefer higher coverage (even with some hallucination risk) while others prioritize safety. The fixed reward can't adapt to these varying preferences.

---

> ### Author Rebuttal · Authors · 2026-03-31
>
> We sincerely appreciate the reviewer's thoughtful comments and the acknowledgment that the problem is "practically relevant and timely" and the ternary scheme is "simple but effective." We address each concern below.
>
>
> **[Q1]: Truthfulness metric uses manually tuned weights. The composite score weights accuracy, abstention, and hallucination rates based on author choices. Different applications might prefer different trade-offs.**
>
> **[A1]**: We clarify that our truthfulness metric follows the standard evaluation practice established in prior works [1,2,3,4,5,6]. That said, we fully agree that different applications may prefer different trade-offs, and we have included a comprehensive ablation study on this factor. In **Table 6**, we compute truthfulness score under three different weight settings and show that TruthRL consistently outperforms all baselines by a large margin across all weight settings, demonstrating TruthRL's robustness across different trade-off preferences.
>
>
> **[Q2]: TruthRL modifies the reward function but relies on existing RL frameworks. Not entirely clear how much gain comes from the ternary reward versus other implementation details.**
>
> **[A2]**: We agree that implementation details can have a significant impact on end performance. Indeed, our experiments directly isolate the contribution of the proposed ternary reward by using the same VeRL framework for training. In **Table 1**, we compare:
>
> - **TruthRL (Binary reward)**: Standard binary reward (+1 for correct, −1 for incorrect), trained with the standard GRPO algorithm, same LLM verifier, training data, hyperparameters, and number of training steps.
> - **TruthRL (Ternary reward)**: Our proposed ternary reward (+1 for correct, 0 for abstain, −1 for hallucinate), with everything else identical.
> This single change (i.e., binary to ternary reward scheme) produces significantly different results and model behaviors (as shown in **Figure 3** and **Figure 5**), clearly demonstrating the contribution of the ternary reward.
>
>
> **[Q3]: Treating abstention as always neutral assumes this is the right default. Some applications may prefer higher coverage (even with some hallucination risk) while others prioritize safety. The fixed reward can't adapt to these varying preferences.**
>
> **[A3]**: We acknowledge that different applications have different requirements, and TruthRL inherently supports this flexibility by design. Specifically, we would like to clarify that:
>
> - **TruthRL supports tunable abstention preference via reward weights**: The ternary reward scheme (correct: +1; abstention: 0; incorrect: -1) is a default configuration, not a hard-coded constraint. Practitioners can straightforwardly adjust these values to match their deployment priorities. For example, in safety-critical applications (e.g., medical QA), one can increase the penalty for hallucination to strongly discourage risky answers. Conversely, in applications where higher coverage is preferred even with some hallucination risk, one can use a less negative reward for hallucination and a more positive reward for correct responses to encourage the model to attempt more answers. This is a simple reward configuration change before RL training. No architectural or algorithmic modifications needed.
>
>
>
> - **Training TruthRL under varying preferences still shows strong performance**: **Table 4** demonstrates that TruthRL accommodates a variety of deployment preferences, including (1) positive abstention reward, (2) neutral abstention reward (default), and (3) negative abstention reward, and achieves strong performance across all settings. As indicated by the results, penalizing abstention increases hallucination while rewarding it encourages excessive abstention, showing the potential of TruthRL adapting to varying application preferences.
>
> Please let us know if you have any further questions, and we are happy to incorporate additional suggestions you might have. If you find our response satisfactory, we would be grateful if you could consider raising your score. Thanks again for your time and efforts!
>
>
> **References**
>
>
> [1]. Yang et al. CRAG -- Comprehensive RAG Benchmark.  NeurIPS 2024.
>
> [2]. Huang et al. ConfQA: Answer Only If You Are Confident. arXiv:2506.07309
>
> [3]. Kachuee et al. PrismRAG: Boosting RAG factuality with distractor resilience and strategized reasoning. EMNLP 2025.
>
> [4] Feng et al. Don’t hallucinate, abstain: Identifying LLM knowledge gaps via Multi-LLM collaboration. ACL 2024.
>
> [5]. Whitehead et al. Reliable visual question answering: Abstain rather than answer incorrectly. ECCV 2022.
>
> [6]. Wen et al. Know Your Limits: A Survey of Abstention in Large Language Models. TACL 2025.

---

> > ### Author Rebuttal · Reviewer_3oXR · 2026-04-03
> >
> > Increasing score from 3 to 4.

---

> > > ### Author Response · Authors · 2026-04-03
> > >
> > > It’s encouraging to know that our response addressed the reviewer’s concerns and led to an increased score. Thank you again for your time and thoughtful comments!

---

### Official Review · Reviewer_WZNa · 2026-03-13

**Soundness:** 2
**Presentation:** 2
**Significance:** 2
**Originality:** 2
**Overall Recommendation:** 4
**Confidence:** 5

**Summary:**

- studies how to train language models to behave more truthfully by encouraging them to abstain when they are uncertain instead of producing hallucinated answers
- authors propose TruthRL, an RL framework that introduces a ternary reward structure that distinguishes between correct answers, abstentions, and incorrect answers. This reward design incentivizes models to answer when they know the solution while avoiding confident hallucinations when they do not
- method is trained using GRPO and evaluated on multiple question answering benchmarks with and without retrieval
- experiments show that the proposed approach reduces hallucination rates and improves overall truthfulness compared to standard prompting, supervised fine-tuning, etc.

**Compliance With Llm Reviewing Policy:**

Affirmed.

**Final Justification:**

While I still have many reservations about the usefulness of this method itself - the discrete ternary reward seems very limited in expressivity, the simple contribution of the ternary reward seems a bit limited in novelty, and complete reliance on a LLM judge seems unideal - I appreciate the authors responding to my concerns, and I will raise my score. New domain results and a greater inclusion of prior work should be included in the revised paper.

**Key Questions For Authors:**

See weaknesses

**Limitations:**

See weaknesses. I am open to raising my score with a much deeper analysis of prior work and a further discussion of the reliance of the LLM verifier.

**Strengths And Weaknesses:**

**Strengths**:
- comparing to a RFT baseline was a very interesting baseline to include
- empirical evaluation is reasonably broad - many diverse datasets, both Llama and Qwen models are tested; baselines include prompting, SFT, RFT, R-Tuning, RLHF, etc.
- main results are strong
- ablations are thorough

**Weaknesses**:
1. Incomplete related work/novelty overstatement. The space this paper claims to fill - that of balancing correctness with calibration - has been addressed by significant prior work (most notably, Beyond Binary Rewards: Training LMs to Reason About Their Uncertainty). Additional works include "SaySelf: Teaching LLMs to Express Confidence with Self-Reflective Rationales" and "Rewarding doubt: A reinforcement learning approach to calibrated confidence expression of large language models". A much deeper discussion of this prior work is needed.

2. Reducing all reward to 3 categories removes information about *degrees* of uncertainty, which removes some expressivity

3. Heavy dependence on LLM verifier: the success of TruthRL seems to rely on strongly on a model-based verifier. In App Table 7, replacing the LLM-based verifier with a rule-based verifier makes the system collapse into overly conservative behavior, with negative truthfulness despite very low hallucination. That means the result is not really “just the ternary reward”; it is “ternary reward plus a strong model judge capable of nuanced labels.” This makes the method more brittle and raises questions about reproducibility, verifier bias, and training signal leakage.

4. Training only done on CRAG; other datasets just used for eval. So it’s unclear if the method still works when trained in other domains or data regimes. Because the paper makes fairly general claims about “developing truthful LLMs,” broader training/evaluation diversity would strengthen the empirical case.

---

> ### Author Rebuttal · Authors · 2026-03-31
>
> We sincerely thank the reviewer for the detailed and constructive feedback. We address each concern below.
>
>
> **[Q1]: A much deeper discussion of this prior work is needed.**
>
> **[A1]**: We appreciate the suggestion and have added an in-depth discussion of these works in our latest revision. To better position our contributions, we also provide a comparison table (Table 1) in this [rebuttal PDF](https://anonymous.4open.science/r/TruthRL-ICML2026-Submission/To_Reviewer_WZNa.pdf), which will also be incorporated in the revision.
>
>
> **[Q2]: Reducing all reward to 3 categories removes information about degrees of uncertainty, which removes some expressivity.**
>
> **[A2]**: While we fully agree that richer reward signals could in principle provide more information, we would like to note that **even though the reward is discrete (ternary), our use of group-relative advantage estimation provides continuous, relative quality information (as defined in Section 3.2)**. For example, an abstention response in a mostly-hallucinating group will receive higher advantage than in a mostly-correct group. The group-relative mechanism thus recovers much of the expressivity that a continuous reward would provide.
>
> That said, we acknowledge that exploring richer reward structures (such as RLCR's Brier score, which provides dense rewards without the need for reward models) is a promising and complementary direction for future work.
>
>
> **[Q3]: In App Table 7, replacing the LLM-based verifier with a rule-based verifier makes the system collapse into overly conservative behavior. This makes the method more brittle and raises questions about reproducibility, verifier bias, and training signal leakage.**
>
> **[A3]**: As noted by the reviewer, **Table 7** highlights that a robust training signal is a prerequisite for RL-based training. This is precisely why we deploy LLM-based verifier instead of a rule-based one in TruthRL.
>
> - **Why rule-based verifiers lead to collapse**: the reason is that exact string matching fundamentally fails to handle semantic equivalence in natural language. In many real-world scenarios, the ground-truth or reference answer is not always a clean few-word string and can take arbitrary forms. For example, the answer "New York City" would be marked incorrect when the reference answer is "The correct answer is NYC.". In such cases, rule-based verifiers can completely distort the training signals, ultimately leading to training collapse.
>
> Regarding the reviewer's specific concerns:
>
> - **Verifier bias**: TruthRL demonstrates strong robustness across multiple independent LLM judges. As shown in **Table 5** of our paper, we evaluate TruthRL using three completely independent judges and TruthRL consistently achieves the highest truthfulness and lowest hallucination across all three evaluators.
>
> - **Reproducibility**: Our LLM-judge setup follows the CRAG benchmark, using exactly the same prompt and pipeline to guarantee consistency. We also provide all code and data in the anonymous repo with detailed instructions for reproduction.
>
> - **Training signal leakage**: we note that the training judge (Llama3.3-70B) is a different model family from our evaluation judges (e.g., Qwen2.5-72B, Gemma3-27B). As shown in **Table 5**, the consistent improvements across different judges strongly argue against training signal leakage.
>
>
> **[Q4]: Broader training/evaluation diversity would strengthen the empirical case.**
>
> **[A4]**: In contrast to training on each evaluation dataset separately, we would like to note that our setup ensures that the model learns to recognize the boundaries of its own knowledge, regardless of the question domain or format. As shown in **Table 1**, TruthRL achieves strong generalization to diverse unseen scenarios: NQ (single-hop factoid), HotpotQA (multi-hop reasoning), and MuSiQue (multi-step compositional), demonstrating that TruthRL learns a general knowledge-boundary recognition capability, not dataset-specific patterns.
>
> Per the reviewer's suggestion, we provide new results on the GSM8K mathematical reasoning benchmark, confirming generalization beyond QA:
>
> | **Method**     | **Truthfulness (↑)** | **Hallucination (↓)** | **Uncertain (-)** | **Accuracy (↑)** |
> |--------------------|:------------:|:-------------:|:---------:|:--------:|
> | Prompting          |     66.7     |     16.3      |    0.7    |   83.0   |
> | SFT                |      -67.4       |      83.7       |   0     |   16.3     |
> | RFT                |      59.8       |     19.6        |   0.9     |  79.5     |
> | R-Tuning           |     -4.0     |      **6.4**      |   91.3    |   2.4    |
> | **TruthRL**            |     **71.9**     |     12.7      |    2.7    | **84.6**    |
>
>
> Please let us know if you have any further questions, and we are happy to incorporate additional suggestions you might have. If you find our response satisfactory, we would be grateful if you could consider raising your score. Thanks again for your time and efforts!

---

> > ### Author Rebuttal · Reviewer_WZNa · 2026-04-03
> >
> > While I still have many reservations about the usefulness of this method itself - the discrete ternary reward seems very limited in expressivity, the contribution of the ternary reward seems a bit limited in novelty, and complete reliance on a LLM judge seems unideal - I appreciate the authors responding to my concerns, and I will raise my score. New domain results and a greater inclusion of prior work should be included in the revised paper.

---

> > > ### Author Response · Authors · 2026-04-03
> > >
> > > Thanks for acknowledging our response and for raising the score! We would like to provide further clarification on each point:
> > > - "Discrete ternary reward seems very limited in expressivity": **As discussed in our response to Q2, GRPO's group-relative advantage mechanism naturally recovers expressivity from the ternary signal**. The discrete reward defines the **ranking** among responses (correct > uncertain > hallucination), while GRPO converts these into continuous, group-normalized advantages. Our ablation (**Table 4**) shows this combination is effective across different weight settings. We also acknowledged that continuous reward signals (e.g., RLCR's Brier score) are a complementary direction, and we discuss this in the revised related work.
> > >
> > > - "The contribution of the ternary reward seems a bit limited in novelty": We respectfully note that the novelty lies not in the ternary reward alone, but in the complete framework. **The unique conceptual advance of TruthRL is the dynamic credit assignment for abstention**: by combining ternary rewards with GRPO's group-relative advantage, the same abstention receives different credit depending on group composition (**Section 3.2**). As shown in **Figure 3**, the ternary scheme produces adaptive abstention dynamics that cannot be reproduced by RL with binary rewards. **Table 1** confirms that the ternary reward consistently outperforms binary variants and other SFT-based or RL-based methods, showing that **TruthRL is a simple yet effective method, as highlighted by Reviewer 3oXR.**
> > >
> > > - "Complete reliance on a LLM judge seems unideal": We would like to clarify that the reliance on an LLM judge is not a requirement of TruthRL itself, but of the application domain. For open-ended QA, semantic equivalence must be assessed, which naturally necessitates an LLM judge as seen across various benchmarks (e.g., CRAG [1], AlpacaEval [2], MT-Bench [3]). Note that **TruthRL can easily be used for verifiable domains without the reliance on LLM as a judge.** For example, when applied to math reasoning, a programmatic judge (e.g., string matching) can be used directly without any LLM judge, as demonstrated in our GSM8K experiments.
> > >
> > > We thank the reviewer again and hope this clarification addresses the remaining concerns. We will ensure to include the new results and expanded related work discussion in our final revision.
> > >
> > >
> > > **References**
> > >
> > > [1]. Yang et al. CRAG -- Comprehensive RAG Benchmark.  NeurIPS 2024.
> > >
> > > [2]. Dubois et al. Length-Controlled AlpacaEval: A Simple Way to Debias Automatic Evaluators. COLM 2024.
> > >
> > > [3]. Zheng et al. Judging LLM-as-a-Judge with MT-Bench and Chatbot Arena. NeurIPS 2023.

---

### Decision · Program_Chairs · 2026-04-30

**Decision:**

Accept (regular)

**Comment:**

This paper was reviewed by four experts in the field.  All reviewers give positive scores. The reviewers indicate that this paper addresses a highly important issue of LLM hallucinations and provides a technically solid and well-motivated approach. The proposed ternary reward is simple but effective, clearly distinguishing hallucinations from abstentions, and demonstrates consistent reduction in hallucination rates while maintaining competitive accuracy across multiple datasets and models. The empirical evaluation is reasonably broad, with strong main results and thorough ablations.
Based on the feedback from reviewers, the decision was made to recommend it for acceptance. We congratulate the authors on their acceptance!